# A polytherapy based approach to combat antimicrobial resistance using cubosomes

Xiangfeng Lai [1], Mei-Ling Han [2], Yue Ding [1,3], Seong Hoong Chow [3], Anton P. Le Brun [4], Chun-Ming Wu[4,5], Phillip J. Bergen[2], Jhih-hang Jiang [2], Hsien-Yi Hsu[6,7], Benjamin W. Muir [8], Jacinta White [8], Jiangning Song [3], Jian Li [2 ✉] & Hsin-Hui Shen [1,3 ✉]

A depleted antimicrobial drug pipeline combined with an increasing prevalence of Gram-negative 'superbugs' has increased interest in nano therapies to treat antibiotic resistance. As cubosomes and polymyxins disrupt the outer membrane of Gram-negative bacteria via different mechanisms, we herein examine the antimicrobial activity of polymyxin-loaded cubosomes and explore an alternative strategy via the polytherapy treatment of pathogens with cubosomes in combination with polymyxin. The polytherapy treatment substantially increases antimicrobial activity compared to polymyxin B-loaded cubosomes or polymyxin and cubosomes alone. Confocal microscopy and neutron reflectometry suggest the superior polytherapy activity is achieved via a two-step process. Firstly, electrostatic interactions between polymyxin and lipid A initially destabilize the outer membrane. Subsequently, an influx of cubosomes results in further membrane disruption via a lipid exchange process. These findings demonstrate that nanoparticle-based polytherapy treatments may potentially serve as improved alternatives to the conventional use of drug-loaded lipid nanoparticles for the treatment of "superbugs".

[1] Department of Materials Science and Engineering, Faculty of Engineering, Monash University, Clayton, VIC 3800, Australia. [2] Infection and Immunity Program, Monash Biomedicine Discovery Institute and Department of Microbiology, Monash University, Clayton, VIC 3800, Australia. [3] Biomedicine Discovery Institute and Department of Biochemistry and Molecular Biology, Monash University, Clayton, VIC 3800, Australia. [4] Australian Centre for Neutron Scattering, Australian Nuclear Science and Technology Organisation, Locked Bag 2001, Kirrawee DC, NSW 2232, Australia. [5] National Synchrotron Radiation Research Center, Hsinchu 30076, Taiwan. [6] School of Energy and Environment & Department of Materials Science and Engineering, City University of Hong Kong, Kowloon Tong, Hong Kong, China. [7] Shenzhen Research Institute of City University of Hong Kong, 518057 Shenzhen, China. [8] CSIRO Manufacturing, Clayton, VIC 3168, Australia. ✉email: Jian.Li@monash.edu; Hsin-Hui.Shen@monash.edu

The World Health Organization (WHO) has declared antimicrobial resistance to be among the top ten global public health threats[1]. Gram-negative bacteria such as *Acinetobacter baumannii*, *Pseudomonas aeruginosa*, and *Klebsiella pneumoniae* are all included in the WHO Priority Pathogens List urgently requiring novel treatment options[2]. A major reason why the aforementioned organisms are so difficult to treat is that they are surrounded by an outer membrane (OM) that acts as an impermeable barrier, preventing antibiotics from reaching their targets inside the bacteria[3,4]. For this reason, polymyxins, available clinically as polymyxin B (PMB) and polymyxin E are considered a last-line therapy for the treatment of antimicrobial-resistant Gram-negative organisms[5,6]. The polymyxins initially bind to lipopolysaccharide (LPS) located within the outer leaflet of OM of Gram-negative bacteria, causing considerable OM disorganization and ultimately cell death[7]. However, reports of polymyxin resistance globally have increased, threatening the utility of these important antibiotics[8,9] and leaving only few new antibiotics currently under development. Therefore, there is an urgent need for novel therapeutic options that may potentially be applicable for use in combating these antimicrobial-resistant Gram-negative "superbugs".

The use of existing polymyxins[10,11] in combination with other antibiotics[12–16] or nonantibiotic adjuvants[17–20] to improve bacterial killing is a promising strategy[21]. Another approach for overcoming antimicrobial resistance involves loading antimicrobials into nanoparticles which transports the antibiotics to their bacterial target sites[22–27]. While nanoparticles have long been used specifically as antimicrobial carriers, the use of nanoparticle-based carriers in polytherapy treatments with antibiotics in order to overcome antimicrobial resistance has been overlooked. Lyotropic liquid-crystalline nanoparticles such as cubosomes have been used as vehicles to deliver antibiotics into the cell[28–30] and disrupt the membrane integrity of LPS-deficient bacteria (*A. baumannii* 19606R and *A. baumannii* 5075D) by solubilizing the OM[31]. In this study, we hypothesized that the combination of cubosomes with an antibiotic such as PMB which also destabilizes the OM might serve as a new strategy to combat antimicrobial-resistant bacteria.

We selected four key antibiotics that are commonly used against Gram-negative bacteria, but with different modes of action (amikacin, aztreonam, doripenem and PMB) for investigation with *A. baumannii*, *P. aeruginosa*, and *K. pneumoniae* (Fig. 1a). The in vitro results showed that only membrane-targeting PMB in polytherapy with cubosomes enhanced bacterial killing against each of the examined pathogens. Subsequently, the antibacterial activity of PMB-loaded cubosomes was compared with the polytherapy treatment of PMB and cubosomes (Fig. 1b), showing that the polytherapy clinically relevant concentrations of PMB and cubosomes is superior in bacterial killing to PMB-loaded cubosomes. Confocal microscopy and neutron reflectometry (NR) were subsequently employed to investigate the detailed interactions of PMB/cubosomes with a model bacterial OM and thus determine the mechanism underpinning enhanced antibacterial activity. To the best of our knowledge, our study is the first to investigate the effect on bacterial killing via polytherapy with an antibiotic and a lyotropic liquid-crystalline lipid-based nanoparticle carrier against Gram-negative pathogens. Our findings offer novel insights that will assist in the design of new nanoparticles and therapies to act as potential adjuvants with existing and emerging antibiotics.

## Results and discussion
### Characterization of cubosomes
Dynamic light scattering (DLS) showed that cubosomes, Octadecyl Rhodamine B Chloride (R18)-loaded and 1–20 wt% PMB-loaded cubosomes had a size range of 150–170 nm and a zeta potential from $-23.8 \pm 1.0$ mV to $1.6 \pm 0.2$ mV (Supplementary Fig. 1 and Supplementary Table 1). SAXS results show that a 1 wt% PMB loading in the cubosomes was found to be biphasic ($Q_{II}^D + Q_{II}^G$ cubic phases; Supplementary Figs. 2 and 3), while 2–20 wt% PMB-loadings resulted in a pure $Q_{II}^G$ cubic phase, suggesting that increasing the amount of positively charged PMB within the cubosomes facilitates a transition from the $Q_{II}^D$ to the $Q_{II}^G$ phase. It is noteworthy that in previous studies, the addition of additives such as charged lipids or nanoparticles also led to the stabilization of the $Q_{II}^G$ phase in excess water[29,32–35]. In addition, the cubosome unit cell parameter and water channel radius expansion increased from 6.51 to 11.12 nm, and from 1.15 to 1.36 nm, respectively, which is driven by an increased electrostatic repulsion of the charged PMB within the cubic-phase nanoparticles (Supplementary Table 2). Cryogenic transmission electron microscopy also revealed the internally ordered structures of the PMB-loaded cubosomes (Supplementary Fig. 4).

The PMB entrapment efficiency (EE%) in 1–20 wt% PMB-loaded cubosomes was found to be higher than 94 % (Supplementary Table 3), indicating that most of the PMB was entrapped within the cubosomes. In vitro release profiles of the PMB-loaded cubosomes (Supplementary Fig. 5) showed that the rate and extent of PMB release increased with PMB concentration. Cubosomes loaded with 1, 2, and 4 wt% PMB released PMB slowly and steadily across the first 6 days, with a cumulative PMB release at this time of 37%, 49%, and 58%, respectively; no further PMB release occurred after this time. Increasing the PMB loading by 8, 16, and 20 wt%, results in a faster release across the first ~18 h, followed by a more gradual release prior to a plateauing. With these higher loading amounts, the cumulative release of PMB was 74%, 85%, and 97%, respectively. The initial faster release of PMB could be due to the rigid $Q_{II}^G$ cubic phase containing less aqueous channels than the $Q_{II}^D$ cubic phase[36], as well as electrostatic repulsion between the charged PMB molecules acting as a driving force for diffusion.

### In vitro antimicrobial properties of the polytherapy treatment of antibiotic-cubosomes and PMB-loaded cubosomes
The minimum inhibitory concentration (MIC) was used to evaluate the antibacterial activity of amikacin, aztreonam, doripenem, and PMB, alone against representative isolates of *A. baumannii*, *P. aeruginosa*, and *K. pneumoniae* in cation-adjusted Mueller–Hinton broth (CaMHB) medium (Table 1). The cubosomes alone were ineffective against all bacterial strains (MICs > 32 µg/mL). The synergistic bacterial killing was measured via fractional inhibitory concentration (FIC) studies in CaMHB (Fig. 2a, c–i, black and red filled bars). The addition of cubosomes did not enhance the efficacy of amikacin, aztreonam, or doripenem (FIC index (FICI) > 1) in all seven strains. Importantly, however, cubosome polytherapy with the LPS targeting antibiotic, PMB, resulted in synergistic bactericidal activity against four of seven isolates (FICI ≤ 0.5) and partial synergy against the other three isolates (0.5 < FICI ≤ 1.0).

We also explored the use of the traditional approach, namely the drug-loaded system, PMB-loaded cubosomes, against the same Gram-negative bacteria cohort (Fig. 2b, c–i, black and red pattern bars). Bacterial killing with all PMB-loaded cubosomes (1% to 20 wt% PMB) was substantially reduced against six of seven isolates when compared to the polytherapy approach as evidenced by the higher MICs (Fig. 2, black pattern bars). In contrast, the PMB-loaded cubosomes slightly lowered the concentration of PMB needed to inhibit the growth of *A. baumannii* FADDI-AB241, as evidenced by their lower MICs, but

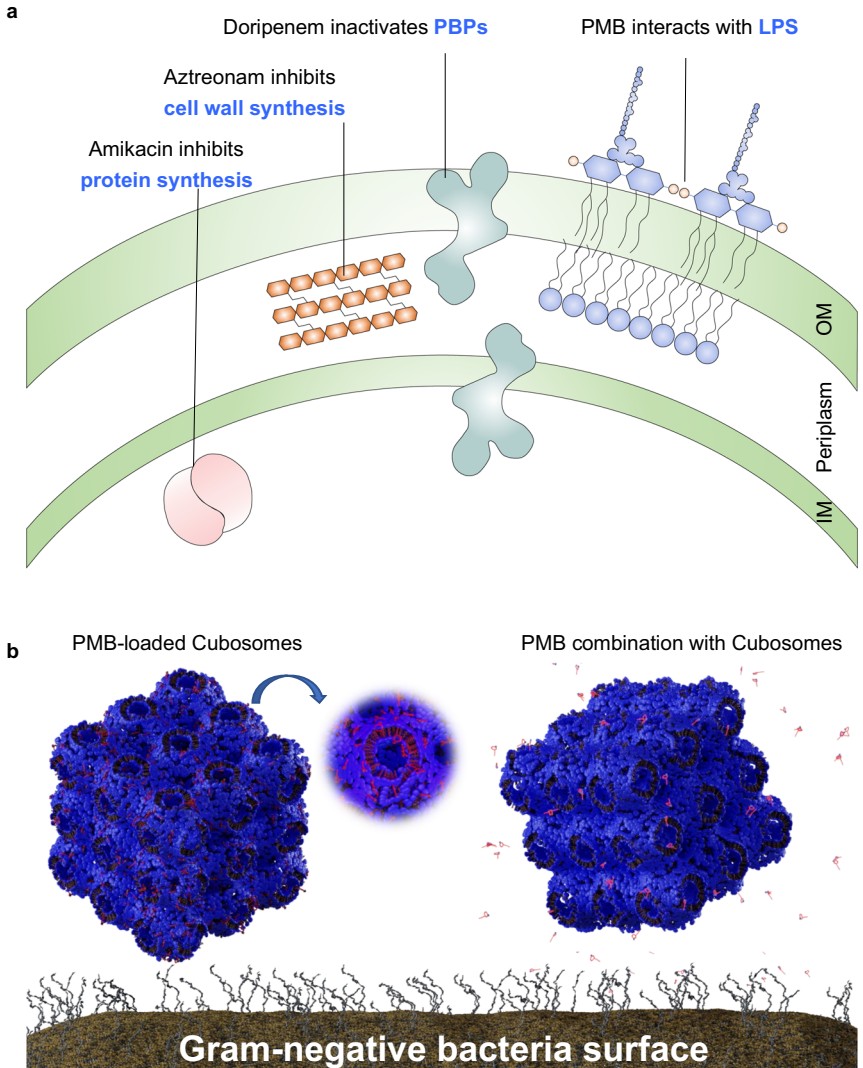

**Fig. 1 The interaction of antibiotics and cubosomes with Gram-negative bacteria. a** Schematic representation of the OM bilayer exposed to different antibiotics which are commonly used in the treatment of Gram-negative bacteria. Amikacin irreversibly binds to the 30S subunit of bacterial ribosomes, interfering with mRNA binding and tRNA acceptor sites, thereby blocking protein synthesis and inhibiting bacterial growth; Aztreonam inhibits the synthesis of the bacterial cell wall by blocking peptidoglycan crosslinking. Doripenem inactivates PBPs by forming stable acyl-enzymes, resulting in a weakened cell wall; PMB interacts with lipopolysaccharide (LPS) of the OM of Gram-negative bacteria, causing the leakage of cytoplasmic content and finally cell death. **b** Schematic diagram of PMB-loaded cubosomes and the polytherapy treatment interacting with the Gram-negative bacterial OM. OM outer membrane, IM inner membrane, PBPs penicillin-binding proteins, mRNA messenger ribonucleic acid, tRNA transfer ribonucleic acid, PMB polymyxin B.

at the cost of using a high concentration of cubosomes (>32 μg/mL, Fig. 2f, red pattern bars). It is worth noting that 20 wt% PMB-loaded cubosomes with a significantly higher concentration of PMB in the system (~3.78 mg loading PMB, Supplementary Table 3) did not enhance the killing effect compared to the polytherapy approach. We further tested the bacterial killing with the polytherapy of PMB with 8 wt% PMB-loaded cubosomes (Supplementary Table 4). The analogous FIC values showed that PMB with 8 wt% PMB-loaded cubosomes did not significantly enhance their activities compared to the polytherapy approach (PMB with cubosomes) although there was more PMB in the bulk solution. These data suggest that the antimicrobials activity is not a PMB concentration dependant effect. The existence of cubosomes in polytherapy plays a crucial role in bacterial killing.

In order to mimic the in vivo conditions, MIC and FIC experiments were also performed in Dulbecco's modified Eagle's medium (DMEM) supplemented with 10 v/v % fetal bovine serum (FBS) (Supplementary Data 1), which is a defined medium for mammalian cells. These results indicated that PMB-loaded cubosomes were more effective against the corresponding bacteria in DMEM/10% FBS than in CaMHB, yet still less potent compared to PMB/cubosome polytherapy in the same culture medium. In addition, the cell viability of a HEK-293T cell line following the addition of polytherapy indicated that there is no appreciable cytotoxicity on the studied mammalian cells up to 32 μg/mL (Supplementary Fig. 6). Notably, recent studies have shown for phytantriol cubosomes with a similar toxicity profile in vitro they were well tolerated in vivo[37–40], which further highlights the potential for use of the polytherapy in vivo. In short, PMB-loaded cubosomes were less active than the polytherapy treatment in inhibiting the growth of Gram-negative pathogens. These data suggest that for lyotropic liquid-crystalline nanoparticles a "one-size-fits-all" approach,

**Table 1 Antimicrobial activity of antibiotics and cubosomes against isolates of *A. baumannii*, *P. aeruginosa*, and *K. pneumoniae* in CaMHB.**

| Bacteria | MIC[a] (µg/mL) | | | | |
|---|---|---|---|---|---|
| | Amikacin | Aztreonam | Doripenem | PMB[b] | Cubosomes[c] |
| *A. baumannii* ATCC 19606 | 26.67 ± 7.54 | 32.00[d] | 2.33 ± 1.25 | 0.67 ± 0.24 | >32 |
| *A. baumannii* 5075 | >32 | >32 | 8.00[d] | 2.00[d] | >32 |
| *A. baumannii* FADDI-AB240 | >32 | >32 | 26.67 ± 7.54 | 1.00[d] | >32 |
| *A. baumannii* FADDI-AB241 | >32 | >32 | 32.00[d] | 26.67 ± 7.54 | >32 |
| *P. aeruginosa* FADDI-PA070 | >32 | 32.00[d] | 6.67 ± 1.89 | 32.00[d] | >32 |
| *K. pneumoniae* ATCC 700721 | 32.00[d] | >32 | 0.33 ± 0.12 | 0.50[d] | >32 |
| *K. pneumoniae* FADDI-KP012 | 18.67 ± 9.98 | >32 | >32 | 26.67 ± 7.54 | >32 |

[a]Minimum inhibitory concentration (MIC) is defined as the lowest drug concentration that prevents visible bacterial growth. The European Committee of Antimicrobial Susceptibility Testing (EUCAST) breakpoints were used to determine susceptibility and resistance to a particular antibiotic (see "Methods" for further clarification).
[b]Resistance for free PMB was defined as an MIC of >2 µg/mL against all bacterial species tested[58].
[c]Resistance to cubosomes was defined as an MIC >32 µg/mL as phytantriol cubosomes exhibited no cytotoxicity up to 50 µg/mL[67].
[d]MIC values were identical across all experiments.
All data were expressed as mean ± SD. All experiments were performed in triplicate ($n = 3$). Mean values and error bars were defined as the mean and SD, respectively. PMB, polymyxin B; CaMHB, cation-adjusted Mueller–Hinton broth. Names of all bacterial taxa are printed in italics.

i.e., focusing only on conventional drug-loaded nanoparticles, is not sufficient for the identification of optimal regimens that maximize bacterial killing.

**Interaction of cubosomes with bacteria visualized by confocal microscopy**. Confocal microscopy was conducted in order to visualize the interactions of cubosome formulations with the bacteria. *A. baumannii* ATCC 19606 was selected as a suitable representative of the Gram-negative bacteria under investigation and treated with 32 µg/mL unloaded cubosomes, 1, 2, 4, 8, 16, and 20 wt% PMB-loaded cubosomes (corresponding to 0.32, 0.64, 1.28, 2.56, 5.12, and 6.4 µg/mL of PMB, respectively, with 32 µg/mL of cubosomes, and a polytherapy combination (0.5 µg/mL of PMB and 32 µg/mL of cubosomes).

When treated with unloaded cubosomes (Fig. 3a), a small number of cubosomes (in red) bound to the cell membrane (in green). This suggests that despite their negative charges (−23.8 ± 1.0 mV, Supplementary Table 1), cubosomes can still interact with the negatively charged bacterial OM (typically −130 to −150 mV)[41]. The previous cell mimetic studies of LPS-deficient model cell membranes have demonstrated that a diffuse cubosome layer forms only on the surface of Gram-negative bacteria without penetrating into the bacterial cells OM[31]. The lack of observed cubosomes penetrating the membrane likely explains their ineffectiveness at killing all the bacterial isolates studied (MICs > 32 µg/mL; Table 1) when used as a monotherapy alone.

The binding of PMB-loaded cubosomes to the bacterial cell membrane was similar to that in the absence of PMB loading (Fig. 3b–d and Supplementary Fig. 7), despite the presence of positively charged PMB in the cubosomes. Surprisingly, a majority of cells following polytherapy treatment had yellow fluorescence after incubation, indicating increased binding of cubosomes to the outside of the bacterial membrane compared to that of the unloaded- and PMB-loaded cubosomes (Fig. 3e). Significantly, PMB is well-known for its ability to interact electrostatically with, and subsequently disrupt, the OM of Gram-negative bacteria[42], while cubosomes are known to cause structural rearrangement of the inner leaflet of the OM[31]. Thus, the enhanced antibacterial activity observed with the polytherapy approach (Fig. 2) is likely due to synergistic interactions of both PMB and the cubosomes with the bacterial OM. To understand the possible mechanisms of these interactions, we employed the use of NR on a model bacterial cell membrane which is reported next.

**Interactions of a polytherapy treatment with a model Gram-negative bacterial OM via NR**. NR is a state-of-the-art tool for non-destructively probing a model bacterial membrane and discriminating between different membrane components with a spatial resolution approaching a few Ångström[43]. Importantly, we have previously demonstrated using NR that cubosomes alone bind only weakly to both a model Gram-negative bacterial OM composed of Ra-LPS, which contains the lipid A and core oligosaccharide region of LPS, and a model inner membrane, but were unable to penetrate into the center of either membrane[31]. By excluding the Ra-LPS and inner membrane, we specifically focused on the hydrophobic domain of LPS, i.e., lipid A. NR was undertaken to investigate the interaction between the components of a polytherapy treatment and a model OM of Gram-negative bacteria consisting of lipid A in the outer leaflet and 1,2-dipalmitoyl-$d_{62}$-sn-glycero-3-phosphocholine (d-DPPC) in the inner leaflet (Fig. 4a). Due to the membrane-targeting action of both PMB and cubosomes, each component has the potential to initially interact with the bacterial membrane surface. Thus, to determine the bactericidal mechanism of the polytherapy treatment, two experiments were conducted whereby the interaction sequence between cubosomes and PMB with the bacterial OM were reversed.

**OM of Gram-negative bacteria—cubosome treatment followed by PMB treatment**. The bilayer was constructed using the Langmuir−Blodgett and Langmuir−Schaefer techniques (Supplementary Fig. 8a, b, see "Methods" for further clarification)[44,45] before being subjected to NR measurement (Supplementary Fig. 8c). The NR profiles of the OM bilayer were then obtained as shown in Fig. 4b. Each isotopic contrast that the OM bilayer was measured under was modeled simultaneously to accurately depict the structure of the asymmetric OM (see Supplementary Fig. 9 for further clarification). The lipid volume fractions across the $SiO_2$ surface for the bilayer were calculated to be as high as 95.3 ± 3.0% (Supplementary Table 5), indicative of the successful fabrication of the OM of Gram-negative bacteria.

Following its construction, the model OM was treated with 32 µg/ml cubosomes. A flattened fringe was clearly observed in the reflectivity curve (Fig. 4c, arrow), indicating OM perturbation. In addition, a decreased scattering length density (SLD) in the inner tail region (Fig. 4d, from black to red curves) was observed. The fitted parameters (Supplementary Table 6) confirmed the presence of a diffuse phytantriol layer (thickness of 50.7 ± 0.1 Å) adsorbed to the surface of the lipid A headgroups of the outer

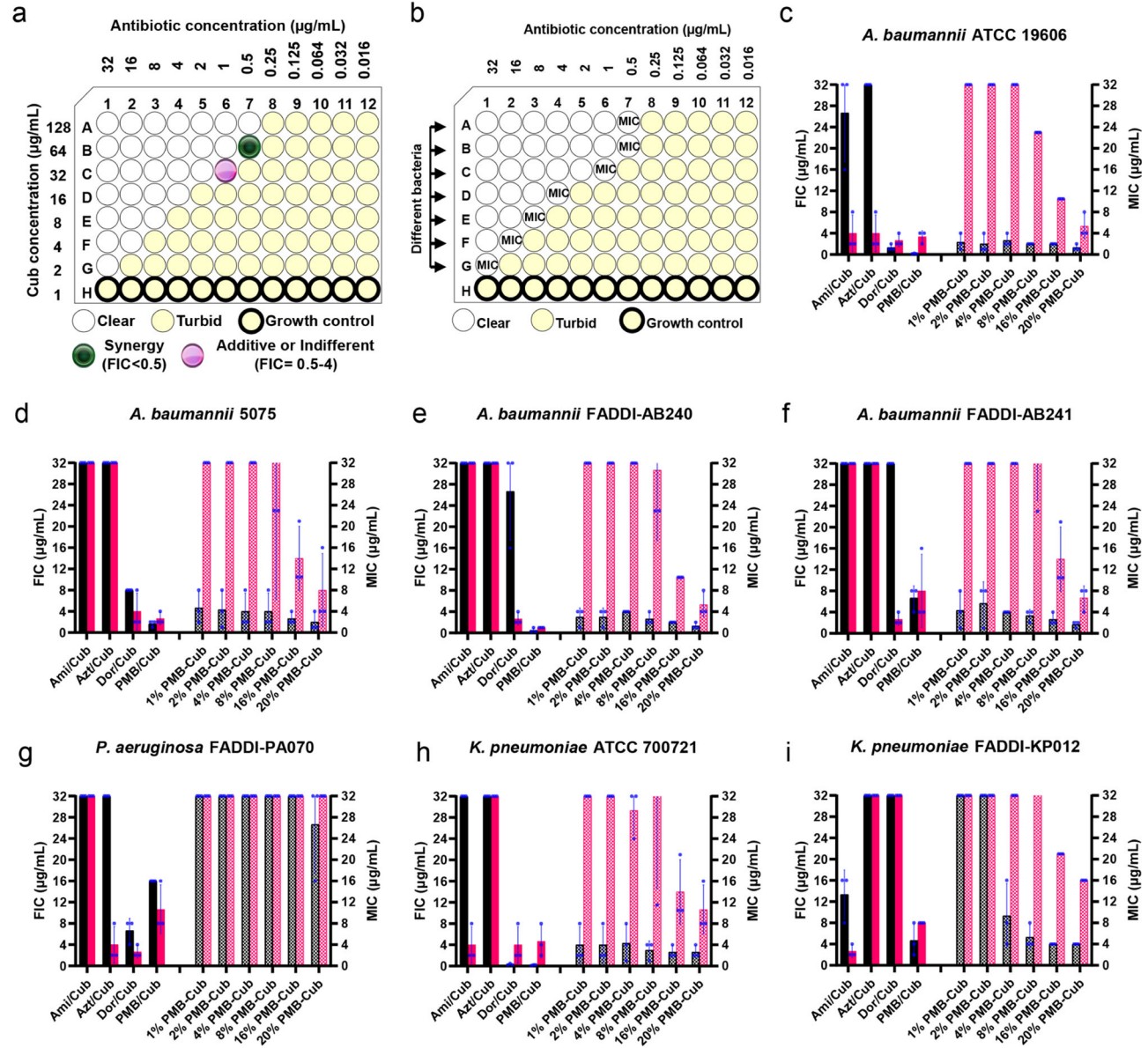

**Fig. 2 Antibacterial activity of the antibiotic/cubosome polytherapy and PMB-loaded cubosome strategies. a** Schematic representation of the FIC synergy checkerboard assay. **b** Schematic representation of the MIC assay. **c–i** FIC of antibiotic/cubosome polytherapy and MIC of (w/w %) PMB-loaded cubosomes against isolates of (**c–f**) *A. baumannii*, (**g**) *P. aeruginosa*, and (**h, i**) *K. pneumoniae* in CaMHB medium. Black and red filled bars represent antibiotics and cubosomes (Cub) in FIC, respectively; Black and red pattern bars represent the loaded PMB and the corresponding Cub in MIC, respectively. All data are expressed as the mean ± SD. All experiments were performed in triplicate (*n* = 3). Mean values and error bars were defined as the mean and SD, respectively. See Source Data for detailed FIC, FICI, and MIC values. MIC minimum inhibitory concentration, FIC fractional inhibitory concentration, Ami amikacin, Azt aztreonam, Dor doripenem, PMB polymyxin B. Source data are provided as a Source Data file.

leaflet. Subsequently, a small volume fraction of phytantriol (~6%, Supplementary Table 6) was detected in the inner and outer leaflets of the OM. The NR results suggest that cubosomes can interact with lipid A and then partly penetrate into the phospholipid layer. However, the bilayer remained highly intact (~81%) as only a small percentage of phytantriol was found in the bilayer.

With the membrane having initially been exposed to the cubosomes, further treatment with 4 μg/mL PMB was undertaken. Following the addition of PMB, there was only a negligible shift in reflectivity profiles (Fig. 4c, from red to blue curve). At the same time, the SLD in both leaflets remained the same (Fig. 4d), implying that the addition of PMB to the cubosome-treated membrane did not induce significant changes to the bilayer structure. The best fit

parameters (see Supplementary Table 7) revealed that PMB was prevented from entering the bilayer as indicated by the absence of PMB in the sublayers. Consequently, the volume fractions of the lipids in the bilayer remained very high (>80.6%). Notably, negligible absorption of PMB or phytantriol cubosomes was observed on the silica surface according to analysis via Quartz crystal microbalance with dissipation (QCM-D) responses (Supplementary Fig. 10)[46]. This suggests that the NR experimental results were not adversely influenced from the silica support. Taken together, NR results suggest that a diffuse cubosome layer accumulated on the membrane but phytantriol did not exchange with the lipids within the bilayers to any significant extent to cause further damage. Further incubation with PMB only led to the PMB adsorbing onto the outer cubosome layer without penetrating into

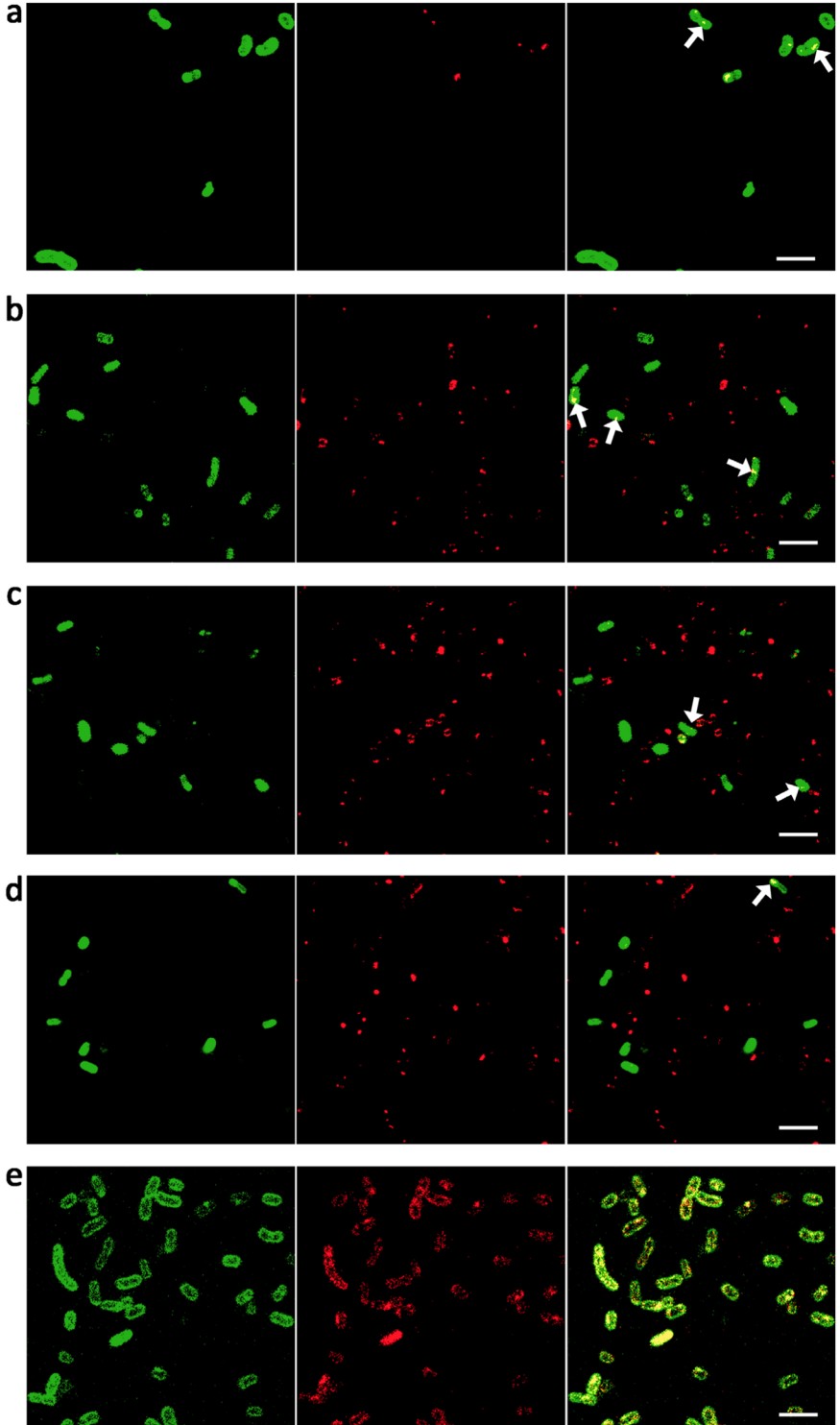

**Fig. 3 Confocal images of viable *A. baumannii* ATCC 19606 labeled with Fix 488 (in green) in the outer membranes.** Cubosomes were labeled with Octadecyl Rhodamine B Chloride (in red), such that the interactions between bacteria and cubosomes would be visible through the colocalization of their individual fluorescence emission in the micrographs. The white arrows indicate the slight interaction of cubosomes or PMB-loaded cubosomes with bacteria. All the images are representative of experiments performed in triplicate (*n* = 3). *A. baumannii* ATCC 19606 treated with (**a**) 32 µg/mL cubosomes alone; **b** 1 wt% PMB-loaded cubosomes; **c** 8 wt% PMB-loaded cubosomes; **d** 20 wt% PMB-loaded cubosomes and **e** polytherapy treatment with 32 µg/mL unloaded cubosomes and 0.5 µg/mL of PMB. Left: bacteria in green; middle: cubosomes in red; right: overlay image. Scale bar: 4 µm.

the membrane structure. Thus, PMB was prevented from reaching the OM due to blockage by the cubosomes which overlaid the OM surface. The lack of membrane damage associated with PMB-loaded cubosomes agrees with the in vitro results (Fig. 2).

**OM of Gram-negative bacteria—PMB treatment followed by cubosome treatment.** Similarly, as to the prior study, the NR data confirmed the presence of the membrane bilayer with a high-volume fraction of lipid of 98.1 ± 3.5% across the SiO₂ surface

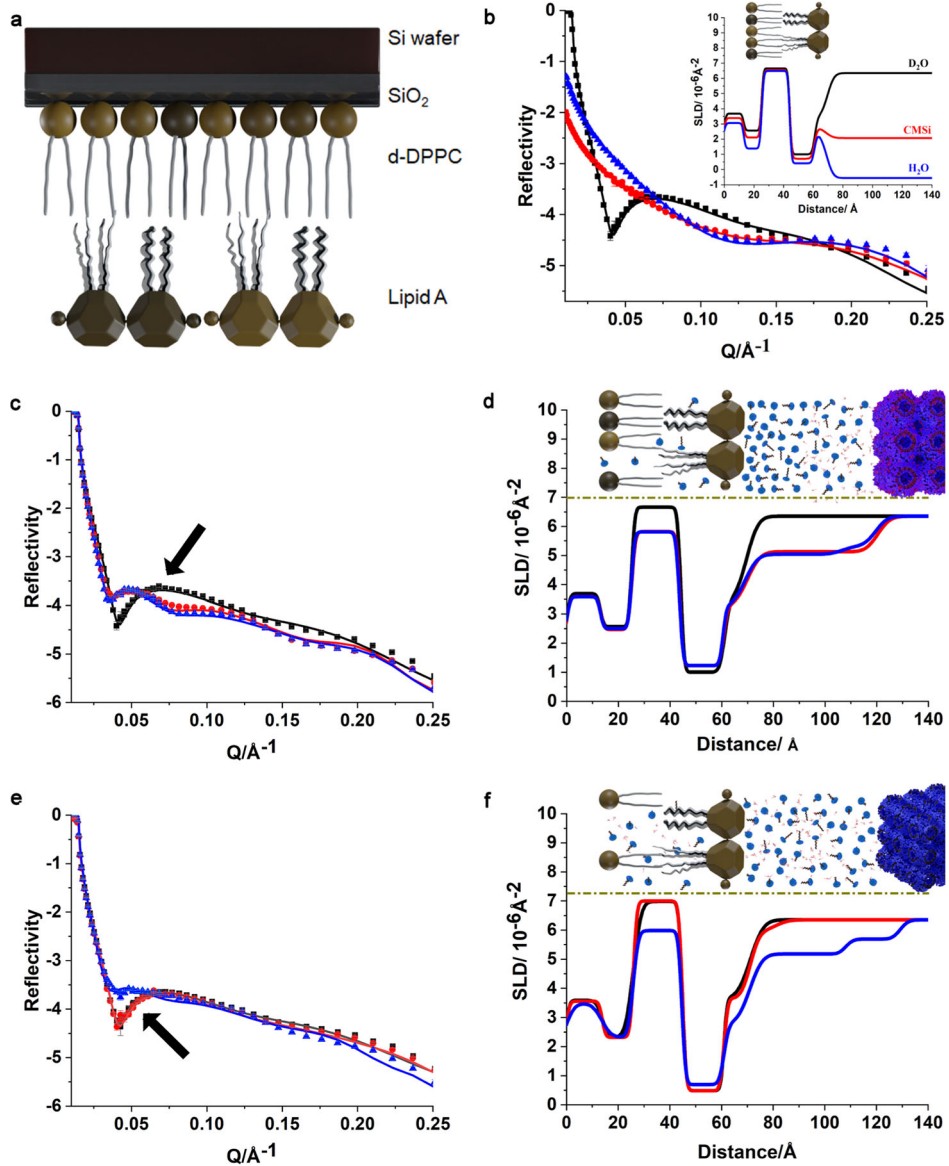

**Fig. 4 Neutron reflectometry (NR) experiment. a** Schematic representation of the model OM bilayer. The lipid A (outer leaflet): d-DPPC (inner leaflet) membrane bilayer are absorbed on the $SiO_2$ surface. During experiments, all space inside the cell is always filled with the aqueous solution, i.e., $D_2O$, CMSi (contrast-matched silicon), and $H_2O$ buffer. **b** The NR profiles (symbols) and fits (lines) of the bilayer obtained from $D_2O$ (black squares and line), CMSi (red circles and line), and $H_2O$ (blue up triangle and line), respectively. The insert is the corresponding scattering length density (SLD) profiles of the bilayer whereby the cartoon depicts the bilayer structure and the distance relative to the $SiO_2$ surface. **c** Experimental (symbols) and fitted (curves) NR profiles of the bilayer (black square and line), treated with 32 µg/mL cubosomes (red circle and line) followed by 4 µg/mL PMB (blue up triangle and line) in $D_2O$; black arrow indicates the fringe changing from black to red curve after incubation with cubosomes. **d** The corresponding SLD profiles of (**c**). The inset depicts the membrane bilayer after successive treatment with cubosomes and PMB. **e** Experimental (symbols) and fitted (curves) NR profiles of the bilayer (black square and line) treated with 4 µg/mL PMB (red circle and line) followed by 32 µg/mL cubosomes (blue up triangle and line) in $D_2O$; black arrow indicates the fringe from black to red curve remained the same after incubation with PMB. **f** The corresponding SLD profiles of (**e**). The insert depicts the bilayer after successive treatment with PMB and cubosomes.

(Supplementary Fig. 11 and Supplementary Table 8). The bilayer was initially treated with 4 µg/mL PMB, followed by a 32 µg/mL cubosome treatment (Supplementary Fig. 11c–f).

In good agreement with our previous study[45], the fringe in the NR curve and the SLD in the tail region remained unchanged (Fig. 4e arrow and Fig. 4f, from black to red curve) following PMB treatment, and a negligible amount of PMB (volume fraction ~2.7–4.8%) was detected in the OM sublayers (Supplementary Table 9). It is thus unlikely that incubation with PMB significantly compromised the membrane structure. We, therefore, propose that PMB primarily binds to the OM surface

without extensive penetration into the lipid bilayer structure at a bulk concentration of 4 µg/mL. Our previous study using a higher concentration of PMB (16 µg/mL) revealed that approximately a volume fraction of 10% of PMB binds to the fatty acyl region of lipid A. This indicates hydrophobic interactions are occurring between PMB and lipid A in addition to the primary electrostatic interactions[44]. The low PMB penetration observed in both experiments is possibly due to the stable gel-like state of the phytantriol bilayer structure at 25 °C. Whereas, at higher temperatures, the bilayer can change phases from a gel to a disordered liquid-crystalline state and enhance PMB

penetration[47,48]. Following subsequent treatment with 32 µg/mL cubosomes, significant changes in the fringe of the NR curve (Fig. 4e, from red to blue curves) and in the tail regions of SLD (Fig. 4f) were observed, indicating solubilization of the bilayer following polytherapy (Fig. 4f, blue curve). Analysis of the NR profiles indicated that large volume fractions of phytantriol (up to $20.3 \pm 0.5\%$) and water (up to $13.4 \pm 6.5\%$) were found in the sublayer (Supplementary Table 10). As a result, PMB with volume fractions ranging from $6.7 \pm 0.9\%$ to $9.7 \pm 0.5\%$ was detected across the bilayer structure, which is possibly due to electrostatic and hydrophobic interactions between the amphipathic PMB and lipids in the sublayers[42,44]. Consequently, a drastic decrease in the volume fraction (from $98.1 \pm 3.5\%$ to $\sim65.6 \pm 0.4\%$) of the bilayer was observed (Supplementary Table 10) which accords with the flattened fringe. Thus, the PMB greatly enhanced the interaction of cubosomes with the model bacterial OM.

**Proposed bactericidal mechanism of the polytherapy treatment of PMB and cubosomes against Gram-negative bacteria.** We have previously reported that cubosomes alone are unable to disrupt the OM of Gram-negative bacteria[31]. Based on the confocal microscopy (Fig. 3) and NR (Fig. 4) results, Fig. 5 illustrates the proposed antimicrobial mechanisms by which the polytherapy treatment disrupted the OM of Gram-negative bacteria. In Fig. 5a, PMB-loaded cubosomes attach to the bacterial membrane surface without penetrating it. The adsorbed layer of cubosomes subsequently prevented the released PMB from binding to the membrane, thus making it more impervious to PMB attack. Consequently, the membrane remained relatively undamaged. Figure 5b describes two distinct modes of action of the polytherapy treatment on the Gram-negative bacterial model OM. First, membrane stability is initially compromised by PMB binding to the membrane surface due to electrostatic interactions between the polycations on PMB and the negatively charged phosphate groups on lipid A[49–51]. Second, the entry of cubosomes into the bilayer is facilitated due to water influx through the partially permeabilized membrane. Cubosomes are then expected to interact with the bilayer lipids via a lipid-exchange mechanism[31], whereby the membrane bilayer is further solvated. This finding indicates that enhanced bacterial killing of Gram-negative bacteria by polytherapy is not contingent upon combining two antimicrobials with separate antimicrobial activity.

In summary, we have demonstrated that cubosomes in a polytherapy or combination approach with the membrane-targeting antibiotic PMB significantly enhance bacterial killing of problematic Gram-negative bacteria, including polymyxin-resistant strains. The PMB-loaded cubosomes, however, did not express a better synergistic effect with PMB compared to cubosomes. Strong interactions between the components during a polytherapy treatment with bacterial OM's were observed by confocal microscopy, yet rarely observed for PMB-loaded cubosomes. Mechanistic studies employing a model cell membrane and NR suggested PMB initially destabilized the Gram-negative OM followed by cubosome-induced membrane solubilization via a lipid-exchange process. These findings pave the way for the future design of nanoparticle-based combination therapies that can potentially overcome Gram-negative bacterial resistance to antimicrobials.

## Methods

**Antibiotics, reagents, and cell line.** Polymyxin B (PMB; lot number 20120204), amikacin (lot number 058K0803), aztreonam (batch number MKCH8931), and doripenem (lot number 0137Y01) were purchased from Sigma-Aldrich and their solutions were prepared in sterilized Milli-Q water (Millipore, Australia). Phytantriol (98%, 3,7,11,15-tetramethylhexadecane-1,2,3-triol), poly(ethylene oxide)-poly(propylene oxide)-poly(ethylene oxide) triblock copolymer (Pluronic F127), lipid A (diphosphoryl from *Escherichia coli* F583), Octadecyl Rhodamine B

Chloride (R18), Dulbecco's modified Eagle's medium (DMEM, 4.5 g/L glucose), fetal bovine serum (FBS) and [3-(4,5-dimethylthiazol-2-yl)-2,5-diphenylte-trazolium bromide (tetrazole)] (MTT) were purchased from Sigma-Aldrich; 1,2-dipalmitoyl-$d_{62}$-$sn$-glycero-3-phosphocholine (d-DPPC) was purchased in powder form from Avanti Polar Lipids Inc. CellBrite$^{TM}$ Fix 488 membrane stain was purchased from Biotium. Milli-Q water 18.2 M$\Omega \cdot$ cm and D$_2$O (99.99 atom%) were used in all experiments for preparation of buffers and solutions. All chemicals were used as received without further purification. The Human embryonic kidney 293T cells (ATCC HEK-293T) was the gift from Dr Thomas Naderer, Monash University.

**Preparation of cubosome formulations.** Eight types of phytantriol based nano-particles were prepared: phytantriol cubosomes; 0.05 wt% R18-99.95 wt% phytantriol (R18-labeled cubosomes); 1 wt% PMB-99 wt% phytantriol (1 wt% PMB-loaded cubosomes); 2 wt% PMB-98 wt% phytantriol (2% PMB-loaded cubosomes); 4 wt% PMB-96 wt% phytantriol (4% PMB-loaded cubosomes); 8 wt% PMB-92 wt% phytantriol (8% PMB-loaded cubosomes); 16 wt% PMB-84 wt% phytantriol (16% PMB-loaded cubosomes); 20 wt% PMB-80 wt% phytantriol (20% PMB-loaded cubosomes); Phytantriol cubosomes were prepared by weighing phytantriol in a glass vial followed by adding a solution of Pluronic F127 in chloroform at a 10 wt% to the phytantriol as previously described[31,52]. The chloroform in the mixture was dried using a nitrogen gas dryer, followed by vacuum drying at room temperature overnight to obtain a viscous and transparent gel. Milli-Q water and/or PBS buffer were then added to the glass vial prior to dispersion by ultrasonication (Qsonica soni-cators, Adelab Scientific) for 5 mins in pulse mode (5 s pulses interrupted by 5 s breaks) at 50% of maximum power (125 Watt, 20 kHz) in ice bath to give a milky dispersion. Finally, the glass vial was sealed and kept at room temperature for further characterization. R18-labeled cubosomes were obtained by adding R18 to the phytantriol prior to Pluronic F127 addition. For PMB-loaded nanoparticles, Pluronic F127 at a 10 wt% of total phytantriol and PMB was added to the phy-tantriol. After overnight drying, the vial was placed in a 70 °C water bath until the gel was melted, 4 mg of PMB dissolved in 50 µL of sterilized Milli-Q water and/or PBS buffer were then added to the vial and thoroughly mixed with the melted gel by vortex and sonication to accomplish a state of homogeneity at room temperature. The final PMB concentration was defined as the actual loaded amount of PMB after removal of the free PMB in the final dispersion.

**In vitro PMB entrapment and release studies.** To separate the free PMB from the cubosome dispersions, ultrafiltration centrifugation was performed prior to the entrapment efficiency determination. 0.5 mL of PMB-loaded cubosomes were fil-tered through an Amicon®Ultra – 0.5 mL centrifugal filter (Merck Millipore Ltd.) with a regenerated cellulose 10,000 Nominal Molecular Weight Limit (NMWL) at 14,000×$g$ for 30 min. The free PMB in the filtrate was collected for further mea-surement. Subsequently, the filters were placed upside down in a fresh Eppendorf tube and centrifuged at 5000×$g$ for 10 min to recover the PMB-loaded cubosomes.

In vitro release profiles of PMB from cubosomes were evaluated using a dynamic dialysis method[53]. Briefly, the recovered PMB-loaded cubosomic dispersions were loaded into a dialysis bag (14,000 MWCO, Millipore, Boston, USA) and dialyzed against 20 mL of phosphate buffer saline (PBS, pH 7.4; the concentration of Na$_2$HPO$_4$, KH$_2$PO$_4$, NaCl, and KCl were 10, 1.8, 137, and 2.7 mmol/L, respectively) thermostatically maintained (37 ± 0.5 °C) and magnetically stirred (200 rpm) throughout the experiment. At 0.5, 1, 2, 4, 18 h, 1d, and then daily up until 12d, 1 mL of solution was removed and replaced by 1 mL of fresh PBS buffer to confirm sink conditions.

Utilizing ultraviolet/visible (UV/Vis) measurement (SparkControl V2.3, Spark 10 M), the amounts of free PMB and the released PMB from the dialysis bag were determined at 210 nm and then equated with standard curve fitting. All the studies and measurements were conducted in triplicate. All the data were processed by Microsoft 365 version 2105 and Originlab 2018 b9.5.1.195.

The percentage of the PMB entrapped in the cubosomes was calculated as: PMB entrapment efficiency (EE%) = $(M_{total} − M_{free})/M_{total} * 100\%$, where $M_{total}$ is the total PMB added in the cubosomes, and $M_{free}$ is the free PMB in the filtrate after ultrafiltration.

**Dynamic light scattering.** The particle size distribution, polydispersity index (PDI) along with the Zeta potential ($\zeta$) of cubosomes were measured on Malvern Zetasizer nanoZS (Zetasizer Software V6.28, Monash Biomedicine Discovery Institute). Samples were diluted with Milli-Q water to a concentration of 0.1 mg/mL and injected into DTS 1070 folded capillary cells. The samples in the cells were stabilized at 37 °C and the DLS was recorded in triplicate at 173° (backscattering angle) for 60 s. $\zeta$ -potentials were recorded 30 runs in triplicate.

**Small-angle X-ray scattering (SAXS).** SAXS was performed at 25 °C using a compact Bruker N8 Horizon instrument (Using software Diffrac. Suite V7.3.1, Monash X-ray Platform, Monash University, Australia), which allows irradiation of samples within a sealed X-ray tube HV generator (Kα radiation from Cu-anode, wavelength $\lambda = 1.5406$ Å) at 50 keV and 1000 µA. The investigated $Q$-range was from 0.007 to 0.387 Å$^{-1}$ (scattering vector Q = [4π sin(θ/2)]/λ, where θ is the scattering angle). Cubosomes at a concentration of 100 mg/mL were then loaded

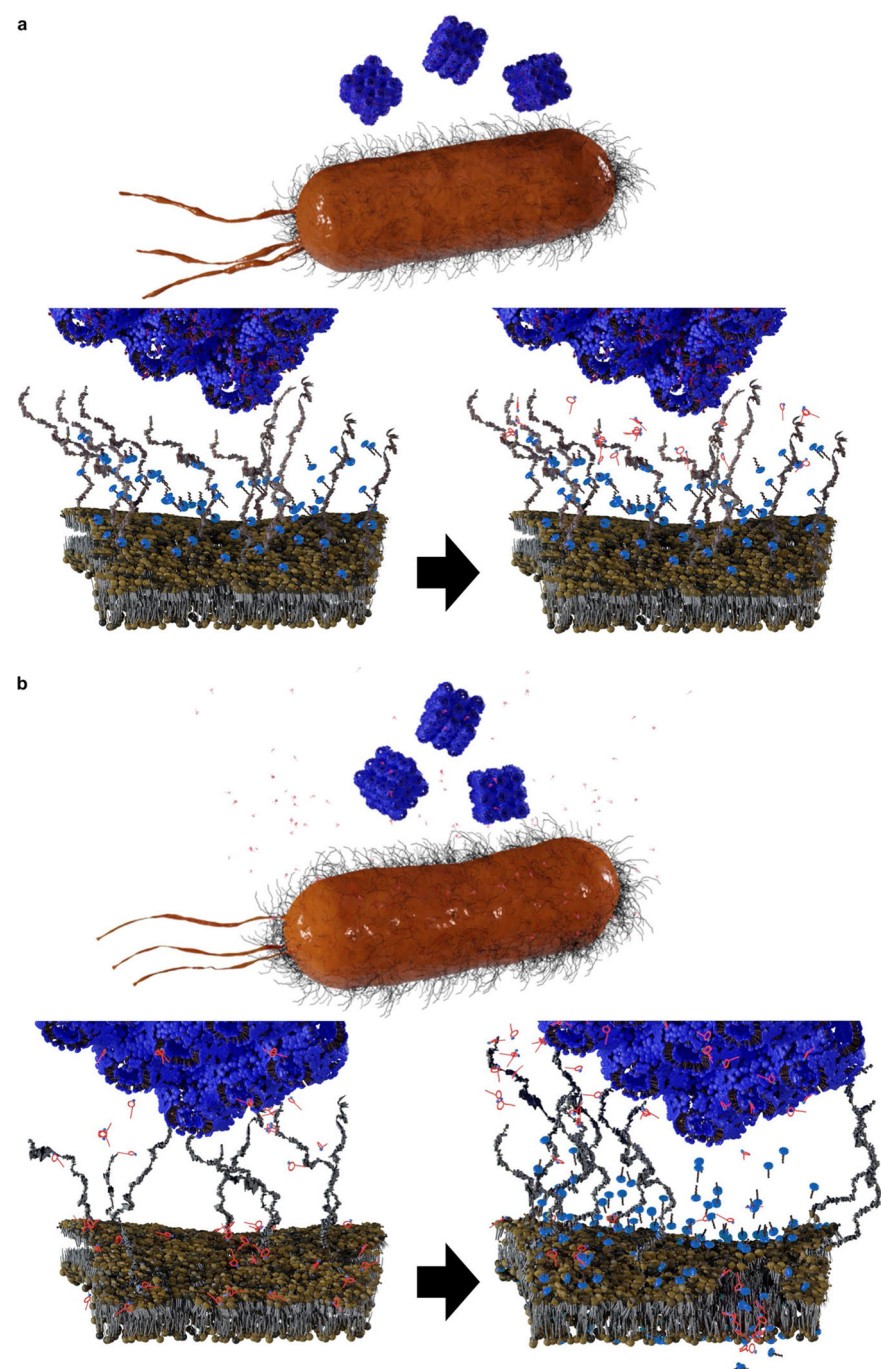

**Fig. 5 A schematic illustration of the possible mechanism of action against Gram-negative bacteria.** Treatment with (**a**) PMB-loaded cubosomes and (**b**) PMB combined with cubosomes. Polytherapy with PMB and cubosomes result in interactions with the bacterial OM in two consecutive ways: PMB initially interacts with the outer leaflet of OM via electrostatic interactions, leading to destabilized areas. Cubosomes then contact with the bilayer, causing further membrane perturbations via a lipid-exchange process.

into a 1 mm (inner diameter) quartz capillary with Milli-Q water as background and Glassy carbon as the reference for determining sample transmission. The sample chamber was kept under vacuum during the measurement. The 2D scattering pattern was then recorded by VÅNTEC-500 Detector for 10 h. The image was then integrated into the 1D scattering function $I(Q)$ and the background of Milli-Q water or PBS buffer in the capillary was subtracted by DIFFRAC.SAXS software (Monash X-ray Platform, Monash University, Australia). A silver behenate standard with a $d$ spacing value of 58.38 Å was used for calibration. The cubic lattice parameter $a$ was calculated as: $a = d_{hkl}\sqrt{(h^2 + k^2 + l^2)}$, where the lattice spacing $d_{hkl} = 2\pi/Q$, $h$, $k$, and $l$ are miller indices. The water channel radius of the cubic structure was estimated by $r_w = [(-\sigma/2\pi\chi)^{1/2}a] - l$, where $\sigma$ and $\chi$ are topological constants ($\sigma = 1.919$ and $\chi = -2$ for $Pn3m$ structure; $\sigma = 3.091$ and

$\chi = -8$ for $Ia3d$ structure), $l$ is the length of the lipid chain (*ca.* 1.4 nm for phytantriol)[54,55]. The estimated $r_w$ equations for cubosomes are given as: $r^{Pn3m}{}_w = 0.391a - 1.4$ and $r^{Ia3d}{}_w = 0.248a - 1.4$.

**Cryogenic transmission electron microscopy (Cryo-TEM).** A laboratory-built humidity-controlled vitrification system was used to prepare the samples for Cryo-TEM. Humidity was kept close to 80% for all experiments, and ambient temperature was 22 °C. In all, 300-mesh copper grids coated with perforated carbon film (Lacey carbon film: ProSciTech #GSCU300FL-50, QLD, Australia) were glow discharged to render them hydrophilic. In total, 3 μl aliquots of the sample were pipetted onto each grid prior to plunging. After 5 s adsorption time the grid was blotted manually using Whatman 541 filter paper, for ~2 s. The grid was then

plunged into liquid ethane cooled by liquid nitrogen. Frozen grids were stored in liquid nitrogen until required. The samples were examined using a Gatan 626 cryoholder (Gatan, Pleasanton, CA, USA) and Tecnai 12 Transmission Electron Microscope (FEI, Eindhoven, The Netherlands) at an operating voltage of 120KV. At all times, low-dose procedures were followed, using an electron dose of 8–10 electrons/Å$^2$ for all imaging. Images were recorded using a FEI Eagle 4k x 4k CCD camera at a range of magnifications using AnalySIS v3.2 camera control software (Olympus).

**Minimum inhibitory concentration (MIC) testing**. The MIC is used to determine the lowest concentration of a drug that prevents the visible growth of a bacterium or bacteria. MIC testing was performed according to Clinical and Laboratory Standards Institute (CLSI) guidelines using amikacin, aztreonam, doripenem, and PMB[56,57]. A colony-forming unit (CFU) is used to estimate the number of viable bacteria in a sample. Briefly, twofold dilutions of the antimicrobial agent were prepared in Cation-Adjusted Mueller–Hinton Broth (CaMHB, 20–25 mg/L of Ca$^{2+}$, 10–12.5 mg/L of Mg$^{2+}$ in Mueller–Hinton Broth, a nutrient-rich bacterial growth medium), or in Dulbecco's modified Eagle's medium (DMEM) containing 10 v/v % fetal bovine serum (FBS). A 96-well microtiter plate was firstly filled with 100 μL of increasing antimicrobial concentrations. The bacterial suspension containing either representative strains of *Acinetobacter baumannii*, *Pseudomonas aeruginosa* or *Klebsiella pneumoniae* was then standardized to 0.5 using a McFarland meter giving ~10$^8$ CFU/mL of the bacteria. The suspension was then diluted 1:100 with CaMHB to achieve ~10$^6$ CFU/mL of the viable bacteria. Within 15 min, 100 μL of the diluted suspensions were vortexed and pipetted into the above wells, resulting the final inoculum of ~5 × 10$^5$ CFU/mL. The MIC was determined after incubation at 37 °C for 16–20 h. The experiment was performed in duplicate for each antibiotic and bacterium. All the data were processed by Microsoft 365 version 2105 and GraphPad Prism version 9.0.1 (151).

Susceptibility and resistance were determined according to the European Committee of Antimicrobial Susceptibility Testing (EUCAST) guidelines[58]. As only breakpoints for colistin (not polymyxin B) against *A. baumannii*, *P. aeruginosa* and *K. pneumonia* have been established by EUCAST and given colistin and polymyxin B have essentially identical in vitro potencies as measured by MICs[59], the EUCAST breakpoints for colistin were applied to polymyxin B. Susceptibility (S) and resistance (R) were defined as MICs ≤2 and >2 μg/mL for polymyxin B, respectively, against all bacterial species tested. For amikacin, S and R were defined as ≤16 and >16 μg/mL against *P. aeruginosa*, and ≤8 and >8 μg/mL for *A. baumannii* and *K. pneumoniae*. For aztreonam, S and R were defined as ≤0.001 and >16 μg/mL for *P. aeruginosa* and ≤1 and >4 μg/mL L for *K. pneumonia*; breakpoints for aztreonam against *A. baumannii* have not been established. For doripenem, S and R were defined as ≤0.001 and >2 μg/mL for *A. baumannii* and *P. aeruginosa*, and ≤1 and >2 μg/mL for *K. pneumonia*.

**Fractional inhibitory concentration (FIC) testing**. FIC is used to determine the impact on the potency of an antibiotic combination in comparison to the individual activities of each antibiotic. FIC testing was undertaken using the microdilution checkerboard technique according to CLSI guidelines[60]. Briefly, twofold dilutions of solutions (four antibiotics, i.e., amikacin, aztreonam, doripenem, and PMB in combination with cubosomes, respectively) were performed in CaMHB. A 96-well microtiter plate was then filled with 50 μL of the appropriate concentrations of antibiotic solution, and this step repeated with cubosome solution (total final volume, 100 μL). A bacterial suspension containing either *A. baumannii*, *P. aeruginosa* or *K. pneumoniae* was then diluted with CaMHB to give ~10$^6$ CFU/mL of final suspension, with 100 μL then dispensed into the wells (final inoculum, ~5 × 10$^5$ CFU/mL). The FIC was determined after incubation at 37 °C for 16–20 h. The experiment was performed in triplicate for each antibiotic and bacterium. All the data were processed by Microsoft 365 version 2105 and GraphPad Prism version 9.0.1 (151).

The FIC index (FICI) is the lowest concentration of antimicrobial agents showing complete inhibition of growth as detected by the unaided eyes. It is calculated by the equation: FICI = (FIC$_{A1}$/MIC$_{A1}$)/(FIC$_{A2}$/MIC$_{A2}$), where A1 = Antibiotic 1, A2 = Antibiotic 2. Synergy was defined as FICI ≤ 0.5, additivity (partial synergy) as 0.5 < FICI ≤ 1.0, indifference as 1.0 < FICI < 4.0, and antagonism as FICI ≥ 4.0[61]. If the endpoint MIC value was unable to be determined due to growth at the highest concentration used, then the next MIC value was used for the calculation (e.g., for an MIC > 32 μg/mL, the MIC used for the calculation was 64 μg/mL)[20].

**Cell viability test**. The cell viability was examined using MTT [3-(4,5-dimethyl-thiazol-2-yl)-2,5-diphenyltetrazolium bromide (tetrazole)] assay. Human embryonic kidney cells (HEK293) were seeded into tissue culture-treated 96-well plates at 5 × 10$^3$ cells per well and incubated at 37 °C in the presence of 5% CO$_2$ overnight. The cells were treated with culture medium containing different concentrations of cubosomes, PMB and PMB/cubosomes overnight. Afterward, the medium was replaced with fresh medium containing 0.5 mg/mL MTT and incubated at 37 °C for 4 h. When finished, the medium was replaced by DMSO and incubated at 37 °C for 10 min. Tecan M200 plate reader (DKSH, Australia) was used to measure the absorbance of the solution at 570 nm.

**Confocal microscopy**. *A. baumannii* ATCC 19606 was used for the microscopy. The bacterium was streaked on CaMHB agar plates and cultured at 37 °C overnight. One colony from the plate was transferred to 10 mL of CaMHB and grown overnight under shaking conditions (200 rpm) at 37 °C. The overnight suspension was then diluted in CaMHB to reach a final concentration of ~10$^8$ CFU/mL as measured by OD$_{600}$ 0.5. Bacterial suspensions (1 mL) were then incubated with the CellBrite$^{TM}$ Fix 488 membrane stain at 37 °C for 15 min and washed thrice with phosphate-buffered saline (PBS). The stained bacterial suspension was centrifuged at 3000×g and the pellets obtained were resuspended in 1 mL of CaMHB. Samples (PMB, R18-labeled cubosomes, R18-labeled PMB-loaded cubosomes) at their MIC values (0.5 μg/mL PMB; for polytherapy, the R18-labeled cubosomes had a fixed concentration of 32 μg/mL; For R18-labeled PMB-loaded cubosomes, 1, 2, 4, 8, 16, and 20 wt% PMB-loaded cubosomes have 0.32, 0.64, 1.28, 2.56, 5.12, and 6.4 μg/mL of PMB, respectively, with 32 μg/mL of cubosomes) were then added to the suspensions respectively and incubated for 2 h at 37 °C under shaking conditions (200 rpm). Bacteria were mounted on a glass microscope slide, anti-fade mounting media, and glass coverslip. Images were acquired using a LEICA-SP5 confocal microscope (Leica Microsystems. Mannheim, Germany) equipped with a ×100 objective (NA: 1.40, oil) lens at Monash MicroImaging, Monash University, Australia. All the data were processed by ImageJ 1.53a.

**Asymmetric bilayer deposition for NR**. Lipid A and d-DPPC were used in this study to mimic the outer and inner leaflets of the Gram-negative outer membrane (OM), respectively. The asymmetric OM bilayer was formed using the Langmuir−Blodgett (LB) and Langmuir−Schaefer (LS) techniques (Supplementary Fig. S1)[31,44,45], whereby a precise control over the components and conformation of each leaflet of lipid bilayer was achieved.

**NR measurement**. NR measurements were conducted on the Platypus time-of-flight neutron reflectometer at the 20 MW OPAL research reactor at the Australian Nuclear Science and Technology Organization (ANSTO), Sydney, Australia[62], using a cold neutron spectrum 2.8 Å ≤ λ ≤ 18 Å. A chopper pairing of choppers 1 and 4 set to a 24 Hz rotation speed was used and provides a wavelength resolution (Δλ/λ) ~8%. Using a vertical scattering geometry, the neutron beams reflected from the sample interface were collected at two glancing angles of incidence (0.85° for 300 s and 3.5° for 3600 s) to cover the momentum transfer (Q) range of interest (0.01 Å$^{-1}$ ≤ Q ≤ 0.3 Å$^{-1}$), where Q is defined as: Q = 4π sin(θ)/λ, where θ is the angle of incidence and λ the wavelength. An illuminated footprint of 33 mm wide and 50 mm long was used.

The asymmetric membrane bilayer was characterized in three isotopic contrasts with contrast-matched scattering length density (SLD, ρ), i.e., D$_2$O (99.9%, ρ = 6.35 × 10$^{-6}$ Å$^{-2}$), CMSi (contrast-matched silicon, 38% D$_2$O: 62% H$_2$O v/v; ρ = 2.07 × 10$^{-6}$ Å$^{-2}$) and H$_2$O (ρ = −0.56 × 10$^{-6}$ Å$^{-2}$). For the exchange between these three contrasts, a total of 7 mL of pH/D 7.4 buffer solution (5 mM CaCl$_2$, 150 mM NaCl, and 10 mM HEPES) was pumped through the sample wafer at the rate of 1.0 mL/min using a HPLC pump (Knauer GmbH, Berlin, Germany). Initially, the bilayer was characterized in buffered isotopic contrasts to confirm the successful formation of the model membrane before incubation for 4 h with the first treatment (32 μg/ml cubosomes or 4 μg/mL PMB), which was subsequently washed out by the injection of D$_2$O buffer to remove any unspecific bindings. The model membrane after the first treatment was then fully characterized under the three isotopic contrasts. Finally, the second treatment (32 μg/ml cubosomes or 4 μg/mL PMB) was added in the same manner to the pre-treated membrane to obtain the final profiles of the model OM of Gram-negative bacteria after polytherapy.

Data reduction was conducted using the SLIM data reduction routine which accounts for detector efficiency, converts the time-of-flight data to wavelength, which is then used to calculate Q, re-bins the data to instrument resolution, stitches the datasets from the two angles of incidence at the overlap region to provide a complete reflectivity profile, and scales the data so that the critical edge is equal to a reflectivity of one.

**NR data analysis**. The analysis of NR profiles was carried out using the MOTOFIT analysis program in Igor pro 6.37 software (WaveMetrics Inc., Portland, OR)[63]. In this study, the bilayer was divided into several sublayers each defined by its thickness, SLD (Supplementary Table S1), and roughness. The reflectometry profiles of the bilayer under different isotopic contrasts (D$_2$O, CMSi, and H$_2$O) were then fitted simultaneously to determine the volume fractions of each component in the sublayers. When a layer is composed of two components, namely a chemical species s and water w, the resultant SLD can be given by: ρ$_{layer}$ = φρ$_s$ + (1 − φ) ρ$_w$, where ρ$_s$ and ρ$_w$ are the SLD of two components (the SLDs used in this study are presented in Supplementary Table 11), respectively, and φ is the volume fraction of chemical species s in the layer. The relative volume fractions of the d-DPPC, lipid A, and PMB in the headgroup of the bilayer structure were unable to be determined because of the minimal isotopic contrast between these three headgroups. Likewise, the relative volume fractions of phytantriol and Lipid A in the tail region were unable to be determined[64]. The volume fractions are shown as a percentage and were determined by the combined volume fractions of the lipid tails region of the bilayer, as the hydration of lipid headgroup regions is more likely to be significantly

higher than that of the tail regions owing to the hydrophilic nature of this moiety of the bilayer[31].

**Quartz crystal microbalance with dissipation monitoring (QCM-D)**. QCM-D is widely used for the evaluation of the binding and interactions between the biological molecules and surfaces within liquid[65]. It was performed by employing Q-SENSE E4 system equipped with an axial flow chamber, IPC High Precision Peristaltic Multi-Channel Dispenser (ISMATEC, flow rate 0.2 mL/min), and 50 nm $SiO_2$-coated quartz crystal sensor[66] at 25 °C. Briefly, the sensors were cleaned by sodium dodecyl sulfate (SDS, 10 mM) for 8 h and then rinsed extensively with Milli-Q water, followed by nitrogen drying and oxidation in UV–ozone chamber for 15 min to remove residual organic impurities. Upon adsorption of materials to the surface of a sensor crystal, the shifts in resonance frequency ($\Delta f$, reflecting the mass of adsorbents, including the coupled water, on the sensor surface) and the changes in energy dissipation ($\Delta d$, correlating with the viscoelastic properties of molecular layers on the surface of the sensor) can be detected from QCM-D using the software QSoft401 V2.5.2.418. The resonance frequency and the dissipation were measured simultaneously at the fundamental natural frequencies 15 and 25 MHz, corresponding to the 3rd and 5th harmonics (overtones) of the 5 MHz crystal. The frequency change ($\Delta f$) of oscillating quartz could be linearly related to its mass change ($\Delta m$) as expressed by the Sauerbrey equation: $\Delta m = -C * 1/n * \Delta f$, where n is the overtone number and C is a constant, which is approximately equal to 17.7 ng/(cm² · Hz) for a 5 MHz AT-cut quartz crystal at room temperature. This means that the addition of 17.7 ng/cm² of mass on a 5 MHz quartz crystal causes a frequency change of 1 Hz. The frequency of 5 MHz quartz can be easily measured with a precision of 0.01 Hz in a vacuum. Therefore, measurement of nanogram-scale masses can be achieved. For example, the corresponding frequency shift on the addition of a monolayer of water with an areal density of ~25 ng/cm² to the surface of an AT-cut quartz crystal is ~1.4 Hz and well within the limits of detection.

**Reporting summary**. Further information on research design is available in the Nature Research Reporting Summary linked to this article.

## Data availability

The authors declare that the data supporting the findings of this study are available within the article and its Supplementary files. Source data are provided with this paper.

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

## Acknowledgements
The authors gratefully acknowledge the use of the facilities at the Monash X-ray Platform and Monash Micro Imaging (Monash University, Victoria, Australia). They thank Heidi Yu, Hasini Wickremasinghe and Zhi Ying Kho for their assistance in the in vitro experiment. The Australian Nuclear Science and Technology Organization (ANSTO) is gratefully acknowledged for providing the neutron research facilities (proposal numbers P6945 and P6451). H.-H.S. is an Australian NHMRC Career Development Research Fellow (GNT1106798) and J.L. is an NHMRC Principal Research Fellow (APP1157909). This work was supported by the NHMRC project grant APP1144652.

## Author contributions
X.L. prepared and characterized the structure of samples, carried out PMB-release and microbiological experiments, and performed parts of the confocal and NR experiments, analyzed all the experimental data, wrote the manuscript with contributions from all of the authors. M.L.H. provided the studied strains and subsequent membrane compositions. Y.D. performed the cell viability assay and part of the NR experiment. S.H.C. carried out part of the confocal microscopy and PMB-release experiment. provided experimental and technical support for the SANS experiments. A.P.L.B. and C.-M.W. provided technical support for NR experiments. P.J.B. revised the manuscript. J.-H.J. provided the support for NR data analysis. H.-Y.H. and J.S. revised the manuscript. B.W.M. and J.W. performed the Cryo-TEM test. J.L. provided the foundation support. H.-H.S. conceived the project, provided the foundation support, and modified the structure of the manuscript.

## Competing interests
The authors declare no competing interests.
