## [Peer Review File · Nature Communications]

Reviewers' comments:

Reviewer #1 (Remarks to the Author):

The article reports the synergistic effect of the membrane active antibiotic PMB with cubosomes formed from lipid/surfactant mixtures. It reports a large amount of characterisation data which shows that whilst PMB alters the structure of the cubosomes it has no effect on their inability to kill GN bacteria. However pre-treatment of the bacteria with PMB, which permeabilises the membranes, allows the cubosomes to have an antibacterial effects. The other attempts at polytherapy were unsuccessful pointing again to the permeabilizing effect of PMB being important. The work does not therefore indicate a wide use of combination therapy with a range of antibiotic types.

The data is clear and the work technically sound. There are a few minor points to address.

It would be helpful to demonstrate whether PMB treatment followed by antibiotic laden cubosomes enhances their activity above that of simpler PMB + antibiotic combination therapy.

The schema in Figure 4F is not easy to understand. The data lines do not suggest that the cubosomes are distanced away from the surface as shown in the cartoon.

The data in 4e shows no effect of PMB on the asymmetric model membrane and it is concluded that PMB does not penetrate the membrane core. The destabilisation is thus driven by surface bound PMB. However, the hydrophobic tail of PMB is crucial for its antibacterial effect and the nonapeptide is many times less active. Thus why should PMB no penetrate the core? It has been shown (Biophys J Vol 88, p1845-1858, 2005) that lipid A -PMB interactions are highly temperature dependent requiring 45 oC for lipid A and 35 oC for LPS, confirmed recently by others using the same methods used here (PNAS 2018 115 (32) E7587-E7594).

Thus is it possible that lack of PMB penetration in the NR results from the model being below the phase transition whilst the MIC was measured at 37 oC. These aspects should be discussed.

Reviewer #2 (Remarks to the Author):

The study by Lai et al. describes the application of phytantriol-based cubosomes with PMB either loaded or as polytherapy to kill Gram-negative bacteria. The work builds on a previous study (Lai et al. ACS Appl. Mater. Interfaces 2020, 12, 44485–44498). The study combines biophysical characterization of the nanoparticles with biological assays.

I have several concerns regarding the results, discussions and conclusions of this study as outlined below. Most conclusions are based on studies with only two repeats and lack of appropriate

statistical analysis. The manuscript further lacks crucial information on sample preparation / experiments that makes it impossible to assess the quality of some data.

Major concerns:

1. Many of the conclusions are based on the comparison of PMB loaded cubosomes with PMB and cubosomes delivered separately. This requires the study of well-defined systems under the same conditions / concentrations. However, the M&M section in the SI lacks information and no experimental evidence is provided that the PMB is actually fully encapsulated in the cubosomes for these samples: The authors state (see experimental section in the SI) that they mix PMB into the phytantriol / F127 gel (bulk) phase and then disperse this gel in water with a tip sonicator (temperature control / energy is not provided). Such a harsh dispersion process will induce phase changes and the release of the PMB into the surrounding water phase. How much of the added PMB is actually still inside the cubosomes and how much is already in the water phase when starting any further experiment then?

This inconsistent loading could also be responsible for the zeta potential data (in the SI), that do not demonstrate a linear change with PMB content, which one would expect.

The release studies (line 99 and SI): E.g. for 1% PMB loading in cubosomes a maximum of 33% release is reported; for 20 % PMB loading the maximum in the plateau is 72% release. Are you sure that you base this on the amount that is actually loaded? Could the initial increase in PMB concentration actually result from unloaded PMB rather than burst release that was observed at high PMB concentration (8, 16 and 20% loading) but not at lower loadings? What was the composition of PBS? Why n only 2?

2. The antimicrobial assays in Figure 2 and related FIC values – error bars are missing. It is not clear if this effect is actually significant or not. A clear and solid statistical analysis should be included. At least 3 repeats would be expected from two independent experiments, and presentation of all data as mean value +/- std. Significance analysis should be included to underline conclusions from the biological assays.

3. Fluorescence microscopy / interaction study: It is not clear if the cubosomes in this experiment are actually still cubosomes! Let me cite from the Figure 3 caption “Cubosomes were labelled with Octadecyl Rhodamine 169 B Chloride (in red),...” It is well known that amphiphilic molecules such as Octadecyl Rhodamine can integrate spontaneously into the internal structure of phytantriol-based cubosomes, leading to composition-induced phase transitions. Did the authors confirm with SAXS that the dye+phytantriol complexes are still cubosomes?

4. Line 147: “This suggests that despite their negative charges (-19.8 ± 1.0 mV, Table S1), cubosomes can still interact with the negatively charged bacterial OM (typically -130 to -150 mV)...”

The authors should consider that Octadecyl Rhodamine 169 B is a positively charged molecule that may integrate into the particle. The zeta potential of this complex should be measured and not that of cubosomes only! These are two different systems. It is not even mentioned how much this dye was added (also missing in the M&M)...

5. Line 204: "The fitted parameters shown in Table S5 confirmed the presence of a cubosome layer (thickness of $50.7 \pm 0.1 \text{ \AA}$) adsorbed to the surface of the lipid A headgroups of the outer leaflet." How can cubosomes with a "average size of 160-170 nm" (line 86) form a 5 nm layer??

6. Error bars and uncertainties are missing at several occasions in the text as well as in Table 1, Figure 2, Figures and Tables in the SI... see comment above.

7. The authors somehow mix the zeta potential with the surface charge which is not the same. Line 87 "Dynamic light scattering (DLS) showed that cubosomes and PMB-loaded cubosomes had an average size of 160-170 nm (Supplementary Fig. S2 and Table S1) and a negatively surface charge (ζ potential) from $-19.8 \pm 1.0 \text{ mV}$ to $-0.9 \pm 0.1 \text{ mV}$." Why is the zeta potential of phytantriol-based cubosomes negative?

8. At some occasions, especially in the abstract and introduction the results are oversold and I would recommend to tone down a bit. E.g. the findings are generalized to Gram-negative bacteria in general. The results section shows that the system is not even highly effective for all the 7 Gram-negative bacteria strains in this study. This should be brought into perspective.

Further concerns:

9. M&M section in the SI, especially also sample preparation info and has to be improved!

To give some examples of missing info: was F127 premixed with water? 50 μL of PMB were added to how much of the gel? Was the gel fully hydrated or only partly? For the PMB concentrations – what is w_{total} ? Does this include water in hydrated cubic bulk phase?

10. For DLS: define size as diameter or radius throughout the ms. What is the "zeta average size" in table S1? This term is new to me.

11. How can the authors measure the zeta potential of PMB in table S1?? This is a molecule that is even too small to measure the Rh with this equipment...

12. SAXS: Description is incomplete. Background subtraction? What is q ? How was the unit cell parameter “ a ” calculated then? Actually, the data in Figure S3 look strange to me for dispersions. The low q upturn that is characteristic for nanoparticles is missing – and intensities appear very high for lab-based experiments on dispersions at 2% mass fraction. Does this figure show SAXS data from the bulk phases?

13. DLS in Figure S2 – what is the “intensity”? Why is 16% profile so different? As on log-scale, the size goes into micron region!? Does Table S1 really summarize these data or is this a different analysis?? The PDI in Table S1 does not represent such a broad distribution presented in Fig S2. Was this a different sample / fitting?

14. For release profiles in Fig. S5: PMB release profile from phytantriol cubosomes (mean \pm SD, $n=2$) ? The related sample preparation is also a miracle to me. In the Figure caption the authors write "2 mg PMB was loaded into cubosomes to make 1, 2, 4 wt % PMB-loaded cubosomes; 3.6 mg PMB was loaded into cubosomes to make 8 wt % PMB-loaded cubosomes; 4 mg PMB was loaded into cubosomes to make 16 and 20 wt % PMB-loaded cubosomes."

This contradicts with description of the method (under SI 1.6) where the authors write “Briefly, PMB-loaded cubosomic dispersions with a total PMB concentration of 4 mg/mL were loaded into a dialysis bag (14,000 MWCO, Millipore, Boston, USA) and dialyzed against 20 mL of phosphate 113 buffer (pH 7.4)...”

15. Antibacterial study: Related data and tables should be explained and presented better with clear description what the abbreviations mean (in addition to statistics to proof if differences are actually significant or not, as mentioned above).

Reviewer #3 (Remarks to the Author):

I found this work interesting, and it is a well-prepared study. However, the concept of combining cubosomes with antimicrobial molecules attacking bacterial membranes has been explored by others, which decreases the novelty.

The work highlights the somewhat urgent need for new antimicrobial active substances in the fight against resistant bacteria. They found that a combination of cubosomes and cationic polymyxins given in a specific order is more active than the mixture nor the substances alone. This is an interesting finding, which is further examined using simplified bacterial model membranes supported on silica surfaces using NR.

My main concerns with the work are the following:

1. It is claimed that this work is the first “To the best of our knowledge, our study is the first to investigate the effect on bacterial killing via polytherapy with an antibiotic and a lyotropic liquid crystalline lipid-based nanoparticle carrier against Gram-negative pathogens”. However, the work recently published by Boge et al. (ACS Appl. Mater. Interfaces 2019, 11, 24, 21314–21322) clearly describes that combination of cubosomes with the antimicrobial peptide LL37 (also targeting the membrane) function as an antimicrobial unit. It is true that they showed that it is the mixture of the two that is active, but I would claim that this is more of a concentration question rather than a mixture vs a specific order question, hence if the authors increase the concentration of PMB in the cubosomes a similar effect could probably be reached as if the PMB is given prior to the cubosomes?

2. It is clearly described in the manuscript that the motivation behind the study is to find out new ways of killing resistant gram-negative bacteria. However, in their studies they do not test their antimicrobials against resistant strains, but instead they evaluate the potency using standard bacterial strains. It would have been much more interesting if the authors would have tested resistant strains.

3. To kill bacteria in vitro is not difficult nor interesting unless the killing substances has low toxicity. I don't see the point of publishing this work unless they perform a proper tox-study to show that these combinations are not toxic to human cells.

4. How well does these combinations work in the presence of blood or serum, or even more important in vivo? Again, killing bacteria in vitro is highly limited.

5. The NR data were performed using model membranes on solid silica supports. The authors should discuss how the presence of the solid support may affect the results. This is of extra importance given that the PMB is positively charged, and the silica surface is highly negative charged (IEP of around 2).

6. More of a detail. The graphical representations based on the NR data show that the inner leaflet of the bilayer is directly in contact with the silica support. It is well known that there is always an entrapped layer of water between the silica support and the bilayer. Why did they not observe this in their models?

Given the overall limitations of the study and relatively low novelty, I do not recommend this work to be published in Nature communications, but rather recommend it to be published in a more specialized journal. However, the need to show tox-data for it to be at all interesting.

Reviewer #1 (Remarks to the Author):

The article reports the synergistic effect of the membrane active antibiotic PMB with cubosomes formed from lipid/surfactant mixtures. It reports a large amount of characterisation data which shows that whilst PMB alters the structure of the cubosomes it has no effect on their inability to kill GN bacteria. However, pre-treatment of the bacteria with PMB, which permeabilises the membranes, allows the cubosomes to have an antibacterial effect. The other attempts at polytherapy were unsuccessful pointing again to the permeabilizing effect of PMB being important. The work does not therefore indicate a wide use of combination therapy with a range of antibiotic types. The data is clear and the work technically sound. There are a few minor points to address.

We are grateful for the reviewer's positive feedback on our works and for the suggestions on improving the current manuscript. Below we will address all of the concerns.

1. It would be helpful to demonstrate whether PMB treatment followed by antibiotic laden cubosomes enhances their activity above that of simpler PMB + antibiotic combination therapy.

The reviewer has raised an interesting question. We therefore conducted this experiment (Table S7, page S23) and the results showed that PMB treatment followed by antibiotic laden cubosomes did not significantly enhance their activity above that of simpler PMB + antibiotic combination therapy. Discussions are shown in manuscript page 7.

2. The schema in Figure 4F is not easy to understand. The data lines do not suggest that the cubosomes are distanced away from the surface as shown in the cartoon.

Thank you for pointing this out. We have now added a diffused phytantriol layer to the cartoon in Figure 4F, page 15.

3. The data in 4e shows no effect of PMB on the asymmetric model membrane and it is concluded that PMB does not penetrate the membrane core. The destabilisation is thus driven by surface bound PMB. However, the hydrophobic tail of PMB is crucial for its antibacterial effect and the nonapeptide is many times less active. Thus, why should PMB not penetrate the core?

We thank the reviewer to point this out. In our study, only a negligible amount of PMB (2.7 ~ 4.8%, at 4 $\mu\text{g}/\text{mL}$ addition) was detected in the membrane. We have adjusted the description accordingly (page 14). Our previous study (published in *ACS Chem. Biol.* 2018, 13 (1), 121-130.) using higher concentration of PMB (16 $\mu\text{g}/\text{mL}$) revealed more PMB penetration in the membrane, indicating hydrophobic interactions are occurring between PMB and membrane. We therefore added more descriptions to clarify this in manuscript page 14 as below:

Additionally, our previous study using a higher concentration of PMB (16 $\mu\text{g}/\text{mL}$) revealed that approximately 10% of the PMB binds to the fatty acyl region of lipid A. This indicates hydrophobic

interactions are occurring between PMB and lipid A in addition to the primary electrostatic interactions.

4. It has been shown (Biophys J Vol 88, p1845-1858, 2005) that lipid A -PMB interactions are highly temperature dependent requiring 45 °C for lipid A and 35 °C for LPS, confirmed recently by others using the same methods used here (PNAS 2018 115 (32) E7587-E7594).

Thus is it possible that lack of PMB penetration in the NR results from the model being below the phase transition whilst the MIC was measured at 37 °C. These aspects should be discussed.

We have cited the above two articles suggested by the reviewer and added the corresponding discussion in manuscript page 14 as below:

The low PMB penetration observed in both experiments is possibly due to the stable gel-like state of the phytantriol bilayer structure at 25 °C. Whereas, at higher temperatures, the bilayer can change phases from a gel to a disordered liquid-crystalline state and enhance PMB penetration.

Reviewer #2 (Remarks to the Author):

The study by Lai et al. describes the application of phytantriol-based cubosomes with PMB either loaded or as polytherapy to kill Gram-negative bacteria. The work builds on a previous study (Lai et al. ACS Appl. Mater. Interfaces 2020, 12, 44485–44498). The study combines biophysical characterization of the nanoparticles with biological assays. I have several concerns regarding the results, discussions and conclusions of this study as outlined below. Most conclusions are based on studies with only two repeats and lack of appropriate statistical analysis. The manuscript further lacks crucial information on sample preparation / experiments that makes it impossible to assess the quality of some data.

We thank the reviewer for the invaluable comments and suggestions. We have done the additional works suggested by the reviewers as summarized below:

1. PMB entrapment efficiency (Table S4, page S19)
2. Provided additional information regarding the sample preparation/experiments (Supplementary section 1.2, page S2).
3. SAXS analysis for the cubosomes in dispersions (Figure S4, page S16).
4. N=3 for MIC, FIC and *in vitro* release studies (Table 1 in page 8, Figure S6 in page S20, Table S5 in page S21, Table S6 in page S22, Table S7 in page S23, Table S8 in page S24).

In summary, the repeated data is highly consistent with our results and has not changed the conclusions of our study.

Major concerns:

1. Many of the conclusions are based on the comparison of PMB loaded cubosomes with PMB and cubosomes delivered separately. This requires the study of well-defined systems under the same conditions / concentrations.

1a. However, the M&M section in the SI lacks information and no experimental evidence is provided that the PMB is actually fully encapsulated in the cubosomes for these samples:

We have further performed the PMB entrapment efficiency measurements and presented the results in Table S4, page S19 shows that more than 94 % of the PMB was loaded into the cubosomes. Corresponding discussion has been added in page 5.

1b. The authors state (see experimental section in the SI) that they mix PMB into the phytantriol / F127 gel (bulk) phase and then disperse this gel in water with a tip sonicator (temperature control / energy is not provided).

We have now included additional experimental information for temperature control/energy in the supplementary section 1.2, page S3.

1c. Such a harsh dispersion process will induce phase changes and the release of the PMB into the surrounding water phase.

We have added the SAXS results regarding the cubosomes' phase after sonication shown in Figure S3-4, page S15-16, indicating they retain stable cubic structure upon sonication.

1d. Such a harsh dispersion process will induce the release of the PMB into the surrounding water phase. How much of the added PMB is actually still inside the cubosomes and how much is already in the water phase when starting any further experiment then?

Please refer to the comment from the Reviewer 2, Q1a

1e. This inconsistent loading could also be responsible for the zeta potential data (in the SI), that do not demonstrate a linear change with PMB content, which one would expect.

The nonlinear change of the zeta potential data was caused by the free PMB in the solution. We have now removed the free PMB in the solution using centrifugal ultrafiltration and have achieved linear changes in the zeta potential as shown in Table S2, page S14.

1f. The release studies (line 99 and SI): E.g. for 1% PMB loading in cubosomes a maximum of 33% release is reported; for 20 % PMB loading the maximum in the plateau is 72% release. Are you sure that you base this on the amount that is actually loaded? Could the initial increase in PMB concentration actually result from unloaded PMB rather than burst release that was observed at high PMB concentration (8, 16 and 20% loading) but not at lower loadings?

We agree with the Reviewer 2. We have now subtracted the free PMB according to the PMB entrapment efficiency (Table S4, page S19). The results are shown in Figure S6, page S20.

1g. What was the composition of PBS? Why n only 2?

We have added the detailed composition of phosphate buffer saline (PBS, pH 7.4; the concentration of Na_2HPO_4 , KH_2PO_4 , NaCl and KCl were 10, 1.8, 137 and 2.7 mmol/L, respectively) in the supplementary section 1.3, page S3. N=3 now.

2. **The antimicrobial assays** in Figure 2 and related FIC values – error bars are missing. It is not clear if this effect is actually significant or not. A clear and solid statistical analysis should be included. At least 3 repeats would be expected from two independent experiments, and presentation of all data as mean value +/- std. Significance analysis should be included to underline conclusions from the biological assays.

We agree that the error bars should be included. The biological results in Table 1 (page 8), Table S2 (page S14), Table S4 (page S19), Figure S6 (page S20), Table S5 (page S21), Table S6 (page S22), Table S7 (page S23), Table S8 (page S24), Figure S7 (page S24) have now been expressed as mean +/- S.D (n=3). Although statistical comparisons between FICs could be performed mathematically, it is rarely used in the field of microbiology and antimicrobial agents. This is because the ranges of FICs and the pharmacological meanings have been defined clearly, i.e. FIC <0.5 (synergy), FIC >4 (antagonism), and 0.5-4 (additive or indifference). We thus feel that the mean +/- std of the FICs from three biological experiments is sufficient to show the consistency between each experiment.

3. **Fluorescence microscopy / interaction study:** It is not clear if the cubosomes in this experiment are actually still cubosomes! Let me cite from the Figure 3 caption “Cubosomes were labelled with Octadecyl Rhodamine 169 B Chloride (in red),...” It is well known that amphiphilic molecules such as Octadecyl Rhodamine can integrate spontaneously into the internal structure of phytantriol-based cubosomes, leading to composition-induced phase transitions. Did the authors confirm with SAXS that the dye+phytantriol complexes are still cubosomes?

We have now included additional SAXS data in Figure S3b in page S15 and Table S3 in page S17 confirming that the dye + phytantriol complexes are still cubosomes with *Pn3m* symmetry.

4. Line 147: “This suggests that despite their negative charges (-19.8 ± 1.0 mV, Table S1), cubosomes can still interact with the negatively charged bacterial OM (typically -130 to -150 mV)...”

The authors should consider that Octadecyl Rhodamine 169 B is a positively charged molecule that may integrate into the particle. The zeta potential of this complex should be measured and not that of cubosomes only! These are two different systems. It is not even mentioned how much this dye was added (also missing in the M&M).

The zeta potential of the complex is listed in Table S2, page S14 indicating that addition of the positively charged dye slightly increased the zeta potential of the cubosomes from -23.8 mV to -19.6 mV. 0.05 wt% of the dye was added into the cubosomes. The descriptions are now added in the supplementary section 1.2, page S2.

5. Line 204: “The fitted parameters shown in Table S5 confirmed the presence of a cubosome layer (thickness of 50.7 ± 0.1 Å) adsorbed to the surface of the lipid A headgroups of the outer leaflet.” How can cubosomes with a “average size of 160-170 nm” (line 86) form a 5 nm layer?? The 5nm layer is a diffused phytantriol layer as suggested by the fitting as shown in Table S10, page S29. We have now reworded the cubosome layer description as a ‘diffused phytantriol layer’ in the manuscript page 13.

6. Error bars and uncertainties are missing at several occasions in the text as well as in Table 1, Figure 2, Figures and Tables in the SI... see comment above.

We have addressed this. Please refer to the comment from the Reviewer 2, Q2.

7. The authors somehow mix the zeta potential with the surface charge which is not the same. Line 87 “Dynamic light scattering (DLS) showed that cubosomes and PMB-loaded cubosomes had an average size of 160-170 nm (Supplementary Fig. S2 and Table S1) and a negatively surface charge (ζ potential) from -19.8 ± 1.0 mV to -0.9 ± 0.1 mV.” Why is the zeta potential of phytantriol-based cubosomes negative?

We agree with the reviewer. We have now reworded the surface charge to zeta potential

The negative zeta potential of phytantriol-based cubosomes could be due to the incorporation of negatively charged Pluronic F127 (as a stabilizer) according to [Express Polymer Letters 10.3 (2016): 216.] A reference has also been cited in Table S2, page S14. It has been widely reported that the zetapotential of phytantriol based cubosomes is negative.

8. At some occasions, especially in the abstract and introduction the results are oversold and I would recommend to tone down a bit. E.g. the findings are generalized to Gram-negative bacteria in general. The results section shows that the system is not even highly effective for all the 7 Gram-negative bacteria strains in this study. This should be brought into perspective.

We agree with the reviewer, and we have toned down the language in the manuscript as suggested by the reviewer.

Further concerns:

9. M&M section in the SI, especially also sample preparation info and has to be improved!

To give some examples of missing info: was F127 premixed with water? 50 uL of PMB were added to how much of the gel? Was the gel fully hydrated or only partly? For the PMB concentrations – what is w_{total} ? Does this include water in hydrated cubic bulk phase?

We agree with the reviewer that more information should be listed in the M&M. We have added additional detailed information regarding the preparation of the samples in Supplementary Information page S2-3, where the percentage of the phytantriol, F127 and PMB are now clearly listed.

10. For DLS: define size as diameter or radius throughout the ms. What is the “zeta average size” in table S1? This term is new to me.

We have defined the size as diameter in Table S2 and reworded the “zeta average size” to particle size (diameter, nm).

11. How can the authors measure the zeta potential of PMB in table S1?? This is a molecule that is even too small to measure the Rh with this equipment...

The reviewer is correct that it is difficult to obtain a reliable size distribution for pure PMB due to its small size and therefore we have opted to remove the zeta potential measurement for the pure PMB.

12. SAXS: Description is incomplete. Background subtraction? What is q? How was the unit cell parameter “a” calculated then? Actually, the data in Figure S3 look strange to me for dispersions. The low q upturn that is characteristic for nanoparticles is missing – and intensities appear very high for lab-based experiments on dispersions at 2% mass fraction. Does this figure show SAXS data from the bulk phases?

We have now added a detailed description for the SAXS analysis in the supplementary information section 1.5, page S4-5, where the background subtraction, q and unit cell parameter were fully described. Yes, the data shown in Figure S3, page S15 were from bulk phases.

13. DLS in Figure S2 – what is the “intensity”? Why is 16% profile so different? As on log-scale, the size goes into micron region!? Does Table S1 really summarize these data or is this a different analysis?? The PDI in Table S1 does not represent such a broad distribution presented in Fig S2. Was this a different sample / fitting?

The intensity distribution is weighted according to the scattering intensity of each particle fraction.

The difference in the 16 wt% profile is referring to the comment from the Reviewer 2, Q1e (free PMB in solution). Yes, Table S1 (now Table S2) summarizes the data from Figure S2 and S4. The broad distribution issue on the 16% profile has now improved after removing free PMB.

14. For release profiles in Fig. S5: PMB release profile from phytantriol cubosomes (mean \pm SD, n=2)? The related sample preparation is also a miracle to me. In the Figure caption the authors write "2 mg PMB was loaded into cubosomes to make 1, 2, 4 wt % PMB-loaded cubosomes; 3.6 mg PMB was loaded into cubosomes to make 8 wt % PMB-loaded cubosomes; 4 mg PMB was

loaded into cubosomes to make 16 and 20 wt % PMB-loaded cubosomes."This contradicts with description of the method (under SI 1.6) where the authors write "Briefly, PMB-loaded cubosomic dispersions with a total PMB concentration of 4 mg/mL were loaded into a dialysis bag (14,000 MWCO, Millipore, Boston, USA) and dialyzed against 20 mL of phosphate 113 buffer (pH 7.4)..."

We thank the reviewer for raising this question. We have conducted the PMB release profile experiments (mean \pm S.D., n=3) and rewritten the description for the preparation of nanoparticles using the exact weight ratio in supplementary information section 1.2, page S2.

15. Antibacterial study: Related data and tables should be explained and presented better with clear description what the abbreviations mean (in addition to statistics to proof if differences are actually significant or not, as mentioned above).

We have carefully checked the manuscript and supplementary information. And we have clearly explained the data and described what the abbreviations mean. For the significance analysis, please refer to the reviewer 2, Q2.

Reviewer #3 (Remarks to the Author):

I found this work interesting, and it is a well-prepared study. However, the concept of combining cubosomes with antimicrobial molecules attacking bacterial membranes has been explored by others, which decreases the novelty. The work highlights the somewhat urgent need for new antimicrobial active substances in the fight against resistant bacteria. They found that a combination of cubosomes and cationic polymyxins given in a specific order is more active than the mixture nor the substances alone. This is an interesting finding, which is further examined using simplified bacterial model membranes supported on silica surfaces using NR. My main concerns with the work are the following:

We appreciate for the reviewer's positive feedback on our work and for the suggestions on improving our manuscript. Below we will address some concerns consecutively.

1a. It is claimed that this work is the first "To the best of our knowledge, our study is the first to investigate the effect on bacterial killing via polytherapy with an antibiotic and a lyotropic liquid crystalline lipid-based nanoparticle carrier against Gram-negative pathogens". However, the work recently published by Boge et al. (*ACS Appl. Mater. Interfaces* 2019, 11, 24, 21314–21322) clearly describes that combination of cubosomes with the antimicrobial peptide LL37 (also targeting the membrane) function as an antimicrobial unit.

We have carefully assessed the publication (Boge et al. *ACS Appl. Mater. Interfaces* 2019, Title: Peptide-loaded cubosomes functioning as an antimicrobial unit against *Escherichia coli*) referred

by the reviewer. However, the authors have never tested the combination of cubosomes with LL37 (cited in manuscript reference 27). The authors have only presented the widely used, traditional approach – using antimicrobial peptide-loaded cubosomes. Therefore, we are confident in claiming that “our study is the first to investigate the effect on bacterial killing via polytherapy with an antibiotic and cubosomes”

1b. It is true that they showed that it is the mixture of the two that is active, but I would claim that this is more of a concentration question rather than a mixture vs a specific order question, hence if the authors increase the concentration of PMB in the cubosomes a similar effect could probably be reached as if the PMB is given prior to the cubosomes?

The reviewer has asked an interesting question. In our current work, we have loaded up to 20 wt% of PMB into the cubosomes (increasing the concentration of PMB in the cubosomes which addresses the queries raised by the reviewer) and the amount of PMB that was actually loaded was calculated to be ~ 3.78 mg (94.4 ± 0.5 % PMB entrapment efficiency in Table S4, page S19). However, the higher loading nanoparticles did not give a similar antimicrobial effect as hypothesised by the reviewer. We have added the corresponding discussion in manuscript page 6.

2. It is clearly described in the manuscript that the motivation behind the study is to find out new ways of killing resistant gram-negative bacteria. However, in their studies they do not test their antimicrobials against resistant strains, but instead they evaluate the potency using standard bacterial strains. It would have been much more interesting if the authors would had tested resistant strains.

We reported data for three PMB-resistant strains: *A. baumannii* FADDI-AB241, *P. aeruginosa* FADDI-PA070 and *K. pneumoniae* FADDI-KP012 in Table 1, page 8. All these three strains are top one priority pathogens according to the World Health Organization guideline which are difficult to treat and can develop resistance.

3. To kill bacteria in vitro is not difficult nor interesting unless the killing substances has low toxicity. I don't see the point of publishing this work unless they perform a proper tox-study to show that these combinations are not toxic to human cells.

We have further evaluated the cytotoxicity of PMB, cubosomes and polytherapy against HEK293 cells. The results shown in supplementary Figure S7, page S25 indicate that polytherapy with a concentration up to 32 µg/mL is non-toxic to mammalian cells.

4. How well does these combinations work in the presence of blood or serum, or even more important in vivo? Again, killing bacteria in vitro is highly limited.

We have added the MIC and FIC results performed in the presence of serum. Results shown in

Table S8, page S24 indicating that polytherapy of PMB and cubosomes exhibited a better antimicrobial activity in the presence of serum compared to those in the absence of serum, which further demonstrates the potential use of the polytherapy *in vivo*. We have added this description to manuscript page 7.

5. The NR data were performed using model membranes on solid silica supports. The authors should discuss how the presence of the solid support may affect the results. This is of extra importance given that the PMB is positively charged, and the silica surface is highly negative charged (IEP of around 2).

We agree with the reviewer. We have further conducted the interaction of PMB and cubosomes with silica surface using QCM-D (Figure S10, page S31). The data suggested that negligible adsorption of PMB at 4 μ g/mL and cubosomes (up to 64 μ g/mL) was observed on the silica support. This suggests that the NR experimental results were not adversely influenced from the silica support. Descriptions have been added in manuscript page 13.

6. More of a detail. The graphical representations based on the NR data show that the inner leaflet of the bilayer is directly in contact with the silica support. It is well known that there is always an entrapped layer of water between the silica support and the bilayer. Why did they not observe this in their models?

Having a thin water layer between the SiO₂ surface and the inner lipid headgroups is possible, however in our case an additional water layer immediately adjacent to the SiO₂ surface did not generate an improvement in the data fitting. Any water later between the SiO₂ and the inner headgroups would be very thin and close to the limit of what the resolution of the NR technique can resolve therefore introducing a water layer could likely introduce artifacts into the fitted results. We have tested our fitting including an additional water layer and the results show that adding an extra water layer did not improve the data fitting. Therefore, we account for this water layer within the inner headgroup layer having a slightly thicker and more hydrated inner headgroup.

REVIEWER COMMENTS

Reviewer #1 (Remarks to the Author):

The authors have responded to my suggestions positively and clearly so I have no further suggestions or requirements

Reviewer #2 (Remarks to the Author):

This is the second time that I am asked to review this manuscript. Parts of it have improved. However, the overall quality is still not very high with several mistakes and inconsistencies in methodology. Its mostly as logical continuation of previous works on lipid nanoparticles for the delivery of antimicrobial drugs / peptides. The technical description is still sub-quality with several mistakes.

Even though the study is about cubosomes (even in the title), the authors base their structural analysis and conclusions on a SAXS study of the phytantriol bulk phase. The bulk phase SAXS data is referred to as from "cubosomes", which is misleading. Cubosomes per definition are particles (somes). The authors tried to address this point in the revision, however this is still not done appropriately.

The authors should be aware that these are different systems and the structure of bulk phases and that their dispersions may be (and are in most cases) very different (e.g., effect of stabilizer, hydration, equilibration, distribution of components between lipid and continuous phases). Not even a note is given on how these bulk phases were prepared for structural analysis. It seems there is also an issue with sample hydration: some of the presented scattering curves show the appearance of an $1a3d$ phase that, to my knowledge, has never been observed under full hydration (i.e., in the dispersion). Further, some of the 2D detector images show orientational effects and micro-crystallization. It appears the bulk samples are not properly equilibrated.

The new SAXS patterns in Figure S4 are also sub-quality. Claiming constancy between bulk and dispersed phase would need a direct comparison of the individual curves with proper indexing of reflections with Miller indices. Why were they measured in MilliQ water and not in PBS buffer as used in the rest of the study? Again, what was subtracted from the curves? Definition of Q also appears wrong.

Reviewer #3 (Remarks to the Author):

I believe that the revised manuscript is much better than the originally submitted version and that the authors have followed all the reviewers comment satisfactory. I recommend that the revised version is to be published in Nature Comm.

Reviewer #2 (Remarks to the Author):

This is the second time that I am asked to review this manuscript. Parts of it have improved. However, the overall quality is still not very high with several mistakes and inconsistencies in methodology. Its mostly as logical continuation of previous works on lipid nanoparticles for the delivery of antimicrobial drugs / peptides. The technical description is still sub-quality with several mistakes.

We are grateful for the reviewer's time and providing feedbacks on our SAXS data in Figure S4 (now Supplementary S2-3) in page S3-4. Below we will address all of the concerns.

Q1. Even though the study is about cubosomes (even in the title), the authors base their structural analysis and conclusions on a SAXS study of the phytantriol bulk phase. The bulk phase SAXS data is referred to as from "cubosomes", which is misleading. Cubosomes per definition are particles (somes). The authors tried to address this point in the revision, however this is still not done appropriately.

We thank the reviewer for pointing this out. To clarify, we did not use bulk phase samples for other experimental characterizations throughout the manuscript. To avoid any misunderstanding to readers of our work, we have removed the SAXS data of the bulk phase (Originally shown in Figure S3, page S15) and have only kept the dispersion data which are always measured on cubosomes as described in the manuscript.

Q2. 2-A. The authors should be aware that these are different systems and the structure of bulk phases and that their dispersions may be (and are in most cases) very different (e.g., effect of stabilizer, hydration, equilibration, distribution of components between lipid and continuous phases). Not even a note is given on how these bulk phases were prepared for structural analysis. Further, some of the 2D detector images show orientational effects and micro-crystallization. It appears the bulk samples are not properly equilibrated.

Please refer to Q1 above. All experimental information is provided to enable any researcher in the field with instructions on how to reproduce our experiments.

2-B. It seems there is also an issue with sample hydration: some of the presented scattering curves show the appearance of an Ia3d phase that, to my knowledge, has never been observed under full hydration (i.e., in the dispersion).

Previous studies suggested that the additives could enable the stabilization of the Ia3d gyroid phase in excess water, as shown in these publications:

In dispersions: ACS Appl. Mater. Interfaces **2016**, 8 (34), 22113-26 & *Colloids Surf. B: Biointerfaces* **2017**, 152, 143-151.

In bulk (in excess water): J. Am. Chem. Soc. **2010**, 132 (47), 16841-16847, *ACS nano* **2015**, 9 (10), 10214-10226 & *Proc. Natl. Acad. Sci. U. S. A.* **2017**, 114 (41), 10834-10839.

We have added a small discussion and these citations to the manuscript highlighted in page 5.

Q3. 3-A. The new SAXS patterns in Figure S4 are also sub-quality. Claiming constancy between bulk and dispersed phase would need a direct comparison of the individual curves with proper indexing of reflections with Miller indices.

We have replotted the SAXS patterns and have added the Miller indices for the reflection in Supplementary Figure 2-3, page S3-4.

3-B. Why were they measured in MilliQ water and not in PBS buffer as used in the rest of the study?

As suggested by the reviewer, we have performed additional SAXS data measurements in PBS as shown below:

- a. 20 wt% PMB-loaded cubosomes in Milli Q water (Supplementary Figure 2, page S3)
- b. 0, 1, 2, 4, 8, 16 and 20 wt % PMB-loaded cubosomes in PBS buffer (Supplementary Figure 3, page S4)

Overall, the corresponding symmetry of the SAXS data in Milli Q water and PBS were the same, where 2-20 wt % PMB-loadings contain Q_{11}^G cubic phase. We have updated our findings in manuscript page 5.

3-C. Again, what was subtracted from the curves? Definition of Q also appears wrong.

We thank the reviewer for pointing this out. The SAXS background has been subtracted from the scattering of MilliQ water or PBS buffer in a capillary. We have added the experimental descriptions to clarify this in SAXS method section in page 22. The scattering vector $Q = [4\pi \sin(\theta/2)]/\lambda$ was also corrected in page 22.

REVIEWERS' COMMENTS

Reviewer #2 (Remarks to the Author):

The authors have addressed my comments and the manuscript increased in quality.

They could still improve the newly added sections by

- considering that two broad Bragg reflections are not sufficient for the assignment of a structure. Can they exclude $Im\bar{3}m$ type? Bonnet ratio?
- using the correct bracket types for Miller indices in peak indexation
- correcting typos.

Reviewer #2 (Remarks to the Author):

We are grateful for the reviewer's positive feedbacks on our revised manuscript. Below we will address all of the concerns.

The authors have addressed my comments and the manuscript increased in quality. They could still improve the newly added sections by:

Q1. Considering that two broad Bragg reflections are not sufficient for the assignment of a structure. Can they exclude $Im3m$ type? Bonnet ratio?

We have added descriptions and references in Supplementary Table 2, page S5 to explain how we confirm $Ia3d$ symmetry of nanoparticles using Bonnet ratio.

Q2. Using the correct bracket types for Miller indices in peak indexation.

Corrected as shown in Supplementary Figures 2-3, page S3-4.

Q3. Correcting typos.

The typos across the manuscript and supplementary document have been carefully checked by two native English speakers: Dr Anton P. Le Brun and Dr Benjamin W. Muir, who are also the co-authors in this manuscript.